# TWIST1 drives endothelial-to-mesenchymal-transition to stabilize atherosclerotic plaques

Blanca Tardajos Ayllon [1,15], Mannekomba Diagbouga [1,15], Ankita Das[1], Siyu Tian[1], Andreas Edsfeldt [2,3,4], Joanna Kalucka [5,6], Jovana Serbanovic-Canic[7], Emily Chambers[7], Jiangming Sun [2], Chrysostomi Gialeli[2], Mark Dunning [7], Sheila E. Francis [7], Xiuying Li[8,9], Akiko Mammoto[10,11], Michael Simons [12], Helle F. Jørgensen [13], Isabel Goncalves[2,3], Suowen Xu [14] & Paul C. Evans [1] ✉

Rupture of unstable atherosclerotic plaques is a major cause of mortality. Endothelial-to-mesenchymal transition associates with advanced atherosclerotic plaques and contributes to plaque progression. We examined the role of *Twist1*, a transcription factor that drives endothelial-to-mesenchymal transition, in plaque progression by inducible deletion from endothelial cells in hypercholesterolemic mice (*Twist1$^{ECKO}$ Apo$^{-/-}$*). Single-cell RNA sequencing coupled to endothelial cell-tracking reveals that *Twist1* promotes endothelial-to-mesenchymal transition in advanced atherosclerotic plaques. Histological analyses demonstrate that endothelial *Twist1* promotes plaque growth and hallmarks of plaque stability (collagen, ACTA2-positive cells) and reduces features of instability (necrosis, macrophage accumulation). Analysis of cultured human aortic endothelial cells shows that TWIST1 contributes to endothelial-to-mesenchymal transition by promoting migration and proliferation through the transcriptional coactivator PELP1. Additionally, TWIST1 promotes endothelial cell proliferation via AEBP1-dependent upregulation of COL4A1. These findings challenge the prevailing view that endothelial-to-mesenchymal transition uniquely destabilizes plaques, by suggesting that TWIST1-driven endothelial-to-mesenchymal transition can promote plaque stability, offering new insights into atherosclerosis pathophysiology and therapeutic potential.

Atherosclerosis, a leading cause of mortality worldwide, is a progressive arterial disease that begins as fatty streaks and advances to complex plaques capable of rupture[1]. Disturbed flow (DF) of blood accelerates the transition of plaques into rupture-prone forms characterized by large necrotic cores, macrophage accumulation, reduced vascular smooth muscle cells, and diminished collagen content[2]. This transformation compromises plaque integrity, elevating the risk of rupture-associated complications such as stroke, unstable angina, and myocardial infarction.

Arterial endothelial cells (ECs), typically organized as a monolayer at the luminal surface, play a critical role in maintaining vascular homeostasis by regulating the influx of inflammatory cells and lipoproteins. However, ECs exhibit phenotypic plasticity, allowing them to adapt their functions and characteristics. One such transformation is

endothelial-to-mesenchymal transition (EndMT), during which ECs lose markers like CDH5 (VE-cadherin) while acquiring mesenchymal markers such as ACTA2 (alpha-smooth muscle actin)[3–5]. EndMT endows ECs with migratory and invasive properties, enabling their movement from the lumen into underlying tissues where they produce collagen and display myofibroblast-like features[4,5]. This process plays a fundamental role in cardiovascular development, including the formation of valve leaflets from subendothelial cushions during embryogenesis[6,7].

EndMT has emerged as a key feature of atherosclerosis, where it is driven by proatherogenic stimuli including DF[8,9] and pro-inflammatory cytokines[10,11]. Single-cell RNA sequencing (scRNA-seq) coupled with cell-tracking technologies has provided compelling evidence of EndMT and other forms of EC plasticity, such as endothelial-to-inflammatory cell transformation (EndIC)[12], in advanced atherosclerotic plaques in mice[10,11,13–17]. Several studies have demonstrated a tight correlation between EndMT and atherosclerosis progression; EC-specific deletion of *Frs2a*[10] and tenascin-X[15] exacerbates EndMT and worsens atherosclerosis, whereas EC-specific deletion of *Shc*[16], *Epsin1*[17], or *IL-1*[11] have the opposite effects. However, the precise role of EndMT in atherosclerosis remains incompletely understood.

EndMT is regulated by multiple transcription factors, including TWIST1, which drives EndMT in contexts such as angiogenesis[18], vascular remodeling[19], pulmonary hypertension[20] and fibrosis[21]. Previously, we demonstrated that TWIST1 promotes lipid accumulation in early atherosclerosis[9]. In this study, we analyse the effects of EC deletion of *Twist1* in pre-existing plaques to further elucidate the role of EndMT in atherosclerosis progression. Our findings reveal that TWIST1 drives EndMT and promotes multiple features of plaque stability. These results challenge the prevailing view that EndMT universally destabilizes plaques, suggesting instead that TWIST1 drives a form of EndMT that may confer beneficial effects in advanced atherosclerosis.

## Results

### TWIST1 expression in murine and human plaque endothelium

scRNA-seq analysis of murine plaques revealed *Twist1* expression in multiple cell types with high levels in EC (Supplementary Fig. 1A). Analysis of human carotid artery plaques revealed that *TWIST1* mRNA levels were enhanced in plaques from asymptomatic compared to symptomatic patients (Supplementary Fig. 1B), and that higher *TWIST1* expression was associated with a reduced risk of future cardiovascular events during post-operative follow-up compared to low levels (Supplementary Fig. 1C). Immunohistochemistry confirmed TWIST1 protein expression in ECs of human carotid plaques (Supplementary Fig. 1D) but this exploratory analysis did not reveal differences between symptomatic and asymptomatic groups. It was concluded that TWIST1 is expressed in plaque endothelium and that its potential role in plaque progression should be investigated.

### Twist1 regulates EC clusters enriched for EndMT markers

To investigate the role of *Twist1* in regulating endothelial phenotypic changes in advanced atherosclerotic plaques, we performed EC-specific genetic deletion of *Twist1* in mice with pre-existing plaques (Supplementary Fig. 2A). This was achieved by feeding *Twist1*[ECKO] *ApoE*[–/–] and control *ApoE*[–/–] mice a Western diet for 14 weeks. Within this period, after 8 weeks, *Twist1* was deleted from the EC of experimental mice using tamoxifen. Quantitative RT-PCR confirmed tamoxifen-induced deletion of *Twist1* in EC of *Twist1*[ECKO] *ApoE*[–/–] mice (Supplementary Fig. 2B).

To examine the impact of *Twist1* deletion on plaque EC phenotypes, scRNA-seq was performed on aortas from control ($N = 5$) and *Twist1*[ECKO] *ApoE*[–/–] ($N = 4$) mice. ECs (CD31 + CD45-) were isolated via enzymatic digestion and FACS sorting before scRNA-seq analysis (Supplementary Fig. 3A). Following quality control filtering, 1173 control and 966 Twist1[ECKO] *ApoE*[–/–] cells were selected for downstream

analysis (Supplementary Fig. 3). Hierarchical clustering analysis identified 11 cell clusters (Fig. 1A). An exploratory analysis of pooled data from *Twist1*[ECKO] *ApoE*[–/–] mice and control mice suggested that clusters 2, 7, and 8 were enriched in cells from control mice, whereas clusters 4 and 5 contained a higher proportion of cells from *Twist1*[ECKO] *ApoE*[–/–] mice mice (Fig. 1B, C, Supplementary Fig. 4A). *Twist1* expression was enriched in cluster 7 and was also prominently detected in clusters 0, 2, 6, and 8 (Supplementary Fig. 5). Overall, these findings suggest that *Twist1* promotes clusters 2, 7 and 8 while suppressing clusters 4 and 5.

Given that TWIST1 is induced by DF and suppressed by uniform flow (UF)[9], we compared the distribution of DF and UF markers among the clusters. Most of the clusters that were promoted by *Twist1* (clusters 2, 7, 8) exhibited an enrichment of DF markers and a corresponding reduction in UF markers, whereas clusters that were suppressed by *Twist1* (clusters 4 and 5) showed the opposite pattern (Supplementary Fig. 6). These findings confirm that *Twist1* is a key regulator of EC responses to DF. We next conducted an unbiased analysis using cluster-specific nested functional enrichment of gene ontology (GO) terms (Supplementary Fig. 7). Cluster 10, which contained the fewest cells, was enriched for pathways related to vascular smooth muscle cell (VSMC) signaling and muscle contractility (Supplementary Fig. 8). As it likely represented contaminating VSMCs, it was excluded from further analysis. *Twist1*-regulated clusters 2, 4, 5, 7 and 8 were associated with diverse EC functions. It was also noticeable that GO terms related to angiogenesis were observed at a higher frequency in *Twist1*-regulated clusters (67%) compared to clusters that were unregulated (0%). Given the close relationship between angiogenesis and EndMT, we next analyzed the distribution of EndMT markers and classical EC markers among clusters (Fig. 1D, Supplementary Fig. 9). EndMT markers were enriched in clusters 2, 7 and 8, which are *Twist1*-regulated, and cluster 3, which is insensitive to *Twist1*. Cluster 8 showed a striking enrichment of EndMT markers alongside a reduction in EC markers, suggesting that this cluster represents an advanced stage of EndMT. In contrast, clusters 2, 3 and 7 exhibited modest alterations in EC markers, suggesting a partial EndMT state. These findings suggest that *Twist1* promotes clusters associated with both advanced (cluster 8) and partial (clusters 2, 7) forms of EndMT in advanced atherosclerotic lesions.

### Twist1 controls EndMT in advanced atherosclerosis

To directly analyse EndMT, we tracked EC-derived cells in *Cdh5*[CreERT2] *Rosa26*[tdTomato] (EC tracking) *ApoE*[–/–] mice that were fed a Western diet for 14 weeks to generate lesions. Tamoxifen was administered at the 8-week time-point to induce EC expression of TdTomato for tracking (Fig. 2A). This was validated by immunofluorescent staining for the endothelial marker vWF, which colocalised with TdTomato in the lumen of the brachiocephalic artery (Supplementary Fig. 10). In control mice, EC-derived cells accumulated within atherosclerotic plaques, accounting for an average of 6% of total plaque cells (Fig. 2B). Among these EC-derived cells, approximately 50% expressed the EndMT markers N-cadherin (NCAD) and SM22α (Fig. 2C). In *Twist1*[ECKO] mice, the accumulation of tdTomato+ EC-derived cells within the plaque was significantly reduced (Fig. 2B), along with decreased proportions of tdTomato+/NCAD+ and tdTomato+/SM22α+ cells (Fig. 2C, D). These findings identify *Twist1* as a regulator of EndMT in atherosclerosis.

### Molecular mechanisms of TWIST1-driven EndMT

To elucidate the molecular mechanisms underlying TWIST1-driven EndMT, we utilized cultured human aortic EC (HAEC) exposed to DF, conditions known to activate TWIST1 both in vitro[9] and in vivo (Supplementary Fig. 6). *TWIST1* silencing was carried out using two different siRNAs; si*TWIST1* (version 1; v1) for initial experiments and si*TWIST1* (v2) for validation experiments. Silencing using *siTWIST1v1* was confirmed (Fig. 3A), and bulk RNA-seq identified 293 genes significantly modulated upon silencing (Supplementary Fig. 11A).

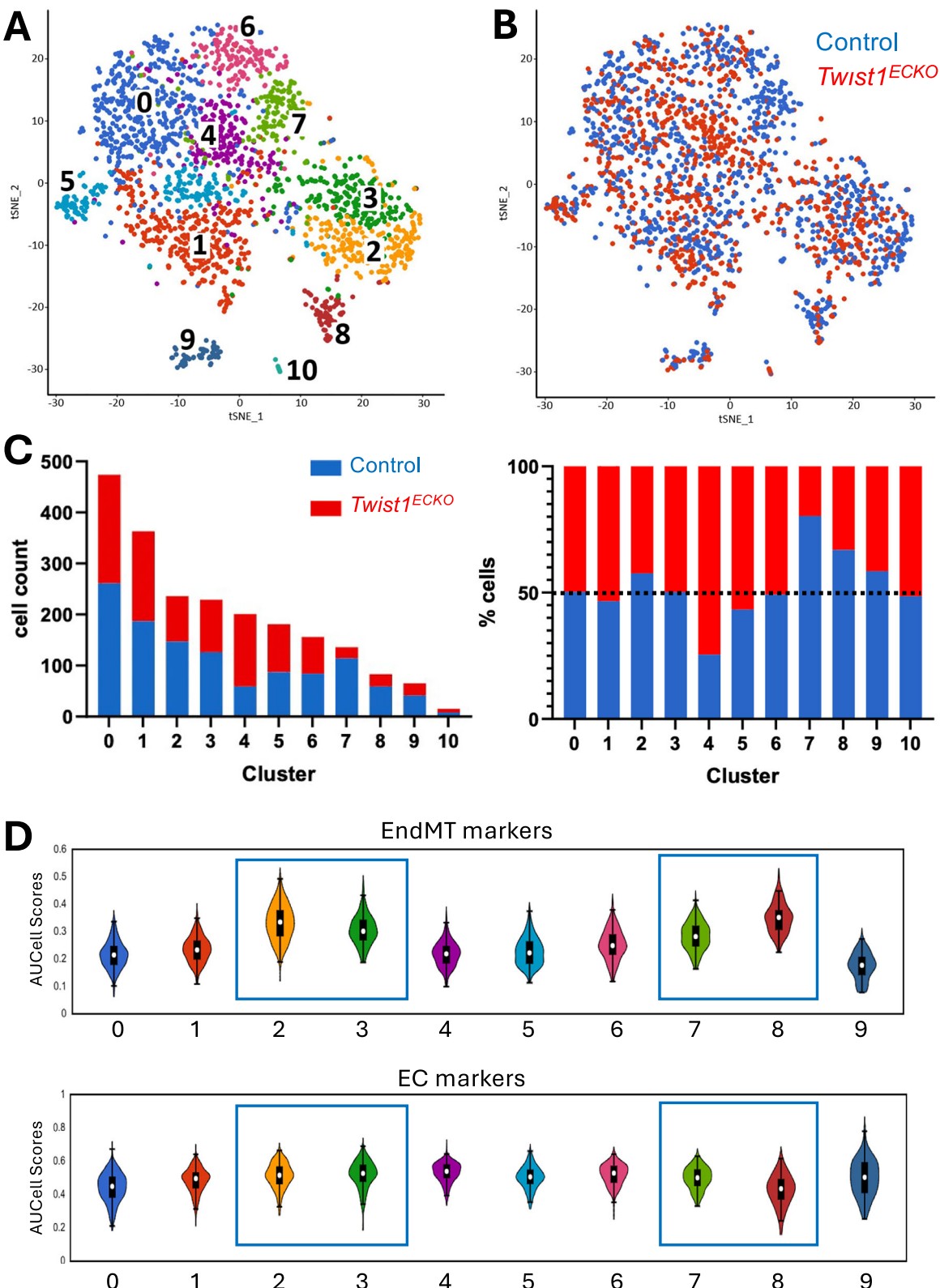

Functional annotation revealed enrichment in gene ontology terms associated with extracellular matrix, development, proliferation, epithelial-to-mesenchymal transition (EMT), and morphogenesis, consistent with a role for TWIST1 in EndMT (Supplementary Fig. 11B). More specifically, si*TWIST1(v1)* silencing significantly downregulated EndMT markers while preserving classical endothelial markers (Fig. 3B, Supplementary Fig. 11C), suggesting a role in partial EndMT. To validate, silencing was carried out using siTWIST1(v2) which reduced

EndMT markers by qRT-PCR (*SNAI1, ACTA2, NCAD;* Fig. 3C) and immunofluorescence (SNAI1, ACTA2; Fig. 3D).

Functionally, si*TWIST1(v1)* silencing reduced cell migration (Fig. 3E, F) and proliferation in HAEC exposed to DF (Fig. 3G) but had minimal effect in HCAECs exposed to UF (Supplementary Fig. 12). These findings demonstrate that TWIST1 orchestrates EndMT in HAEC exposed to DF by promoting migration and proliferation transcriptional programs. We took a structured approach to define the mechanisms underlying

**Fig. 1 | scRNA-seq analysis of *Twist1*-regulated EC clusters.** Male *Twist1*[ECKO] (*Twist1*[fl/fl] *CdhS*[CreERT2/+] *ApoE*[−/−]; n = 4) and control (*Twist1*[fl/fl] *CdhS*[+/+] *ApoE*[−/−]; n = 5) mice aged 8 weeks were fed a Western diet for 8 weeks to induce atherosclerotic lesions. Tamoxifen was then administered for 5 consecutive days to induce *Twist1* deletion in experimental mice, and a Western diet was provided for an additional 6 weeks. Aortas were analysed by FACS of pooled CD31[+] CD45[−] cells coupled to scRNA-seq. **A** t-SNE map of single-cell RNA sequencing from *Twist1*[ECKO] and control mice, colored by cluster assignment. Clusters were identified using unbiased hierarchical clustering. **B** t-SNE map showing the cell contribution of *Twist1*[ECKO] and control mice to each subpopulation. **C** Bar graphs showing the absolute number of cells from *Twist1*[ECKO] and control mice in each cluster (left), and the proportion of *Twist1*[ECKO] and control cells in each cluster (right). The dotted line indicates the 50% threshold. Clusters 2, 7 and 8 are largely composed of ECs derived from control mice, whereas clusters 4 and 5 are mainly composed of ECs derived from *Twist1*[ECKO] mice. **D** Average levels of EndMT markers and EC markers (AUCell score) were calculated for individual cells and are presented as violin plots. The white dot represents the median. The box is the interquartile range (IQR) (the upper limit of the box is quartile 3 (Q3), the lower limit is quartile 1 (Q1)). The thin lines (whiskers) extend to the lower and upper adjacent values (Q1 − 1.5 IQR and Q3 + 1.5 IQR). Clusters 2, 3, 7 and 8, which are highlighted with a blue box, show enriched expression of EndMT markers.

TWIST1-driven migration and proliferation. Firstly, we compared bulk RNA-seq data from HAECs and scRNA-seq data from plaque endothelium and identified 45 genes positively regulated by TWIST1 across both systems (Fig. 3H). From this set, we prioritized 24 genes with known or putative roles in EMT or EndMT from literature searches and used qRT-PCR to confirm *TWIST1*-dependent regulation for 10 of them (Supplementary Fig. 13A). These 10 genes were then subjected to siRNA silencing, which was validated by qRT-PCR (Supplementary Fig. 13B), to assess their potential effects on migration and proliferation. This screening identified a single gene, *PELP1*, that regulated both processes, and seven genes (AEBP1, COL4A1, FKBP10, KDELR3, RPS7, SEC23B and USP14) that specifically regulated proliferation (Supplementary Fig. 13C–E, Figs. 4, 5). We focused subsequent analyses on three genes, *PELP1, AEBP1 and COL4A1*.

Focusing on the transcriptional co-activator PELP1, we confirmed regulation by *TWIST1* using the alternative version *siTWIST1* (v2) (Supplementary Fig. 13F). *PELP1* silencing significantly reduced both cell migration (Fig. 4A–C) and proliferation (Fig. 4D), indicating a critical role in these processes. To elucidate the functional link between PELP1 and *TWIST1*, we reintroduced PELP1 in *TWIST1*-silenced cells via lentiviral overexpression. Silencing using *siTWIST1(v2)* reduced migration (Supplementary Fig. 14) and proliferation (Supplementary Fig. 15), thereby validating *TWIST1* as a positive regulator. PELP1 restoration rescued the defects in migration (Supplementary Fig. 14) and proliferation (Supplementary Fig. 15), demonstrating that PELP1 functions downstream of *TWIST1* to mediate these cellular processes. Consistently, bulk RNA-seq revealed that *PELP1* silencing altered the expression of pathways involved in migration and proliferation (Fig. 4E). Notably, 34% (99/288) of TWIST1-regulated genes were also regulated by PELP1 (Supplementary Fig. 16), consistent with a TWIST1-PELP1 pathway. TWIST1 regulation of PELP1 was further validated at the protein level via immunoblotting (Fig. 4F). To determine whether TWIST1 functions as a potential transcriptional activator of *PELP1*, we analyzed the *PELP1* gene. A putative TWIST1 binding site was identified within a regulatory region marked by H3K27 acetylation (H3K27Ac) (Fig. 4G). ChIP-qPCR on anti-FLAG immunoprecipitates from TWIST1-FLAG-expressing ECs confirmed TWIST1 binding to the PELP1 regulatory region (Fig. 4G), with the SNAI2 promoter as a positive control (Supplementary Fig. 17), supporting its role in transcriptional regulation. We validated the TWIST1-PELP1 axis in vivo. Moreover, EC-deletion of *Twist1* reduced PELP1 expression in tracked EC within murine brachiocephalic plaques (Fig. 4H). Collectively, these findings suggest that TWIST1 promotes migration and proliferation by upregulating PELP1.

Among the genes regulating proliferation, we focused on the transcription factor AEBP1 and COL4A1, hypothesizing an interaction given the role of AEBP1 as a collagen regulator[22]. We confirmed regulation of *AEBP1* and *COL4A1* by *TWIST1* using the alternative version *siTWIST1 (v2)* (Supplementary Fig. 13F). Silencing of AEBP1 or COL4A1 significantly reduced cell proliferation (Fig. 5A). To assess their functional relationship with TWIST1, we reintroduced *AEBP1* or *COL4A1* in *TWIST1*-silenced cells via lentiviral overexpression. Restoration of either gene rescued proliferation (Supplementary

Fig. 15), indicating that AEBP1 and COL4A1 act downstream of TWIST1 to drive this process. Consistently, bulk RNA-seq analysis revealed that AEBP1 silencing affected pathways related to proliferation and matrix remodeling (Fig. 5B). Moreover, comparative RNA-seq analysis identified 23 proliferation modulators co-regulated by *AEBP1* and *TWIST1* (Supplementary Fig. 18), supporting a TWIST1-AEBP1 pathway. TWIST1 regulation of AEBP1 and COL4A1 was validated at the protein level (Supplementary Fig. 19).

Consistent with a transcriptional mechanism, ChIP confirmed TWIST1 binding at potential regulatory sites within the AEBP1 and COL4A1 genes (Fig. 5C, D). Additionally, siRNA silencing demonstrated that AEBP1 regulates COL4A1 expression (Fig. 5E). At a functional level, replenishing collagen type IV in the extracellular matrix rescued the proliferative capacity of *TWIST1*-silenced cells (Fig. 5F), indicating a critical role for collagen type IV in TWIST1-driven proliferation. We validated our findings in vivo by showing that EC-specific *Twist1* deletion reduced AEBP1 (Fig. 5G) and COL4A1 (Fig. 5H) expression in tracked ECs within murine brachiocephalic plaques. Our data support the suggestion that *Twist1* regulates cluster 8, which is enriched for *Aebp1*, *Col4a1* and *Pelp1* (Supplementary Fig. 5). These observations are consistent with a TWIST1−AEBP1−COL4A1 axis that promotes EC proliferation. We conclude that TWIST1 regulates EndMT in atherosclerosis through a PELP1-mediated migration pathway and an AEBP1-COL4A1-driven proliferation.

## *Twist1* promotes features of plaque stability

To investigate the role of *Twist1* in plaque progression, we analysed disease severity after EC-specific genetic deletion of *Twist1* in male mice with pre-existing plaques. *Twist1*[ECKO] *ApoE*[−/−] and control *ApoE*[−/−] mice were fed a Western diet for 14 weeks. Within this period, after 8 weeks, *Twist1* was deleted from the EC of experimental mice using tamoxifen (Fig. 2A). An additional cohort of *ApoE*[−/−] mice was exposed to a Western diet for 8 weeks prior to plaque analysis to provide baseline measurements (Supplementary Fig. 20, Supplementary Fig. 21). *Twist1*[ECKO] *ApoE*[−/−] mice exhibited a significant reduction in plaque size in brachiocephalic arteries (Fig. 6A) and aortic roots (Fig. 7A) compared to controls. Plaque size increased relative to baseline measurements in both *Twist1*[ECKO] *ApoE*[−/−] and control groups (Figs. 6A, 7A). These data suggest that *Twist1*[ECKO] reduces plaque growth but does not cause regression. This reduction was not associated with changes in plasma cholesterol or triglyceride levels, which were similar between experimental and control groups (Supplementary Fig. 22). Collagen and necrotic content of plaques increased relative to baseline measurements in the control group, whereas the proportions of macrophages and ACTA2[+] cells remained relatively stable (compare Fig. 6, Supplementary Fig. 20, Fig. 7 and Supplementary Fig. 21). Notably, *Twist1*[ECKO] *ApoE*[−/−] mice showed reduced fibrillar collagen content and a lower proportion of ACTA2-positive cells, both indicators of plaque stability, in brachiocephalic (Fig. 6B, C) and aortic root (Fig. 7B, C) plaques. In contrast, plaques of *Twist1*[ECKO] *ApoE*[−/−] mice exhibited increased macrophage content (Figs. 6D, 7D), enhanced necrosis (Figs. 6E, 7E) and more elastin breaks (Fig. 6F), all features of plaque instability. These findings support the conclusion

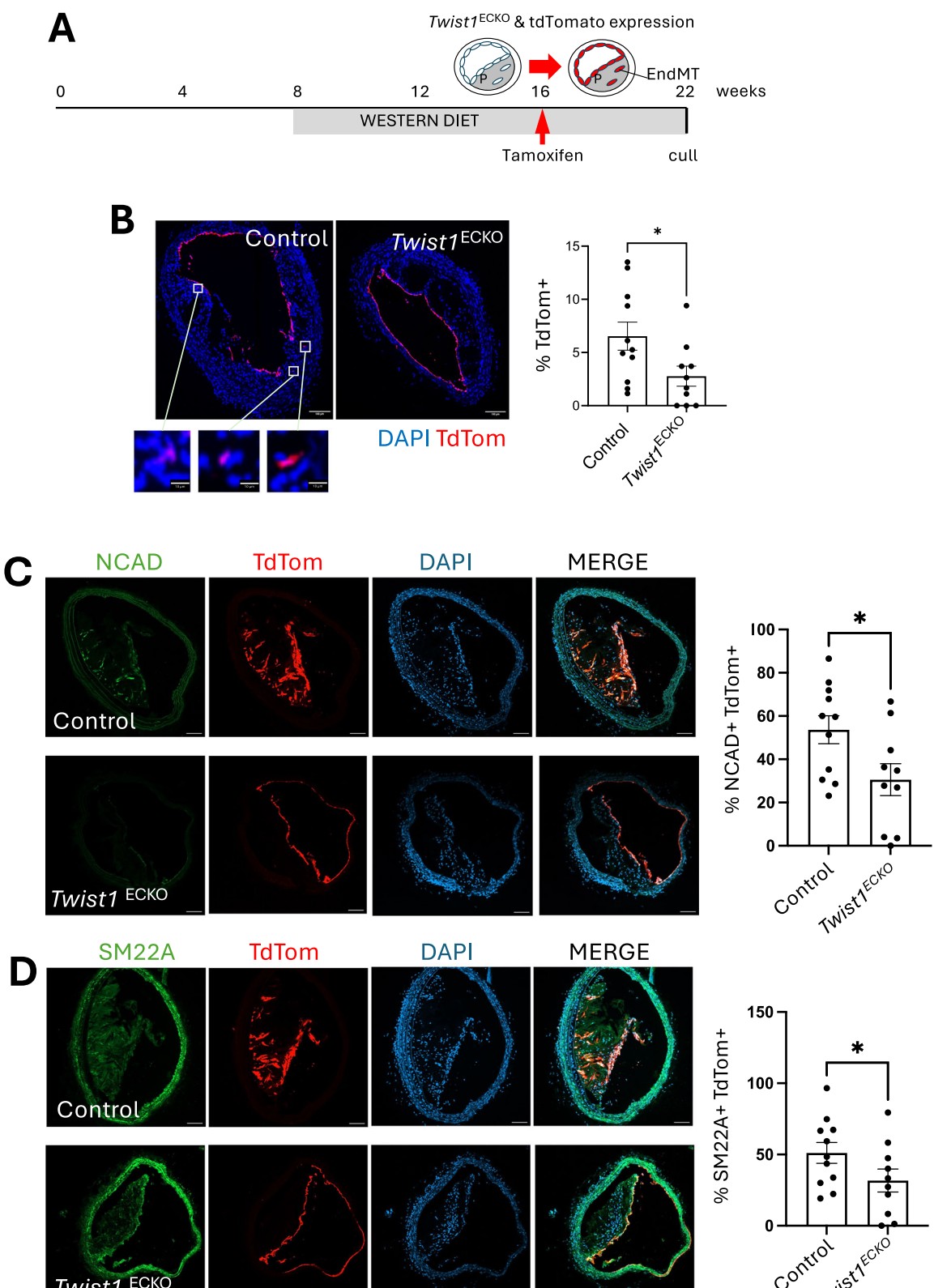

that *Twist1* plays a critical role in promoting plaque growth while also contributing to features of plaque stability.

## Sex differences in *Twist1* control of EndMT and features of stability

We conducted parallel experiments to analyse the influence of *Twist1* on EndMT and plaque stability in female mice. EC tracking demonstrated EndMT in brachiocephalic plaques of female mice (Supplementary Fig. 23A). However, endothelial deletion of *Twist1* did not influence EndMT (Supplementary Fig. 23A), plaque size (Supplementary Fig. 23B), or features of plaque stability (Supplementary Fig. 23C–G) in female mice. These findings indicate that *Twist1* regulation of EndMT and its impact on features of plaque stability are observed in male mice but not in

**Fig. 2 | *Twist1* promotes EndMT in vivo. A** Timeline of *Twist1* deletion on an EC-tracked background in a model of atherosclerotic progression. Male *Twist1*[ECKO] (*Twist1*[fl/fl] *Cdh5*[CreERT2/+] *ApoE*[−/−] *Rosa26*[TdTomato/TdTomato]) and control mice (*Twist1*[+/+] *Cdh5*[CreERT2/+] *ApoE*[−/−] *Rosa26*[TdTomato/TdTomato]) aged 8 weeks were fed a Western diet for 8 weeks to induce atherosclerotic lesions. Tamoxifen was then administered for 5 consecutive days to induce *Twist1* deletion and TdTomato expression in ECs, and a Western diet was provided for an additional 6 weeks (totaling 14 weeks of Western diet). **B** The percentage of Rosa26TdTomato[+] cells within the plaque was quantified in *Twist1*[ECKO] (*n* = 10) and control (*n* = 11) mice. Rosa26TdTomato[+] cells are shown in red, and nuclei are counterstained with DAPI (blue). The magnified view of the boxed region shows Rosa26TdTomato[+] cells in the plaque. Representative images are shown (Scale bar = 100 μm or 10 μm for magnified views). Mean values are shown +/- standard errors. **C**, **D** Frozen sections of brachiocephalic arteries from *Twist1*[ECKO] (*n* = 10) or control (*n* = 11) mice were stained using antibodies against NCAD (**C**; green) or SM22A (**D**; green). Rosa26TdTomato[+] cells are shown in red, and nuclei are counterstained with DAPI (blue). The percentage of NCAD +/ TdTom + cells (**C**) or SM22A / TdTom+ cells (**D**) within the plaque was calculated. Representative images are shown (Scale bar = 100 μm). Mean values are shown +/- standard errors. Differences between means were analysed using a two-sided unpaired *t*-test (**B**) or one-sided unpaired *t*-test (**C**, **D**). * *P* < 0.05. Source data are provided as a Source Data file.

females, highlighting the sex-dependent mechanisms of atherosclerosis.

## Discussion

EndMT contributes to the cellular composition of atherosclerotic plaques[10,11,13–17]. In human atherosclerotic plaques, EndMT marker expression is enhanced in advanced disease compared to earlier stages[10,13]. Studies in hypercholesterolemic mice have shown that modulating EndMT-related pathways influences disease progression[10,11,15–17]. These observations support the concept that EndMT plays a pathogenic role in the progression of atherosclerosis towards unstable rupture-prone lesions. Our findings challenge this perspective, demonstrating that *Twist1* promotes EndMT while increasing plaque stability. Although this conclusion goes against the current paradigm, it is nevertheless intuitive because several canonical features of EndMT, such as collagen deposition and the accumulation of ACTA2-positive cells, are also hallmarks of plaque stability.

To integrate our findings with existing literature, we propose that EndMT in atherosclerotic plaques represents a phenotypic continuum with diverse functional outcomes. Our scRNA-seq analysis supports this interpretation, identifying four clusters enriched in EndMT markers. Among them, Cluster 8 exhibited a more advanced EndMT phenotype, characterized by the downregulation of endothelial markers alongside the upregulation of EndMT markers. In contrast, Clusters 2, 3, and 7 retained endothelial markers alongside EndMT markers, indicative of partial EndMT, which has been previously described in angiogenesis[23,24], pulmonary hypertension[20,25], and post-myocardial infarction remodeling[26]. TWIST1 promoted Clusters 2, 7, and 8 but had no effect on Cluster 3, suggesting that TWIST1 regulates specific subsets of EndMT cells in atherosclerosis. This idea of multiple EndMT subsets aligns with several scRNA-seq analyses that have underscored the phenotypic heterogeneity of ECs across organs and disease states, revealing that EndMT represents a spectrum of phenotypes rather than a singular state[12,27,28]. It also resonates with observations of EMT, where epithelial cells transition into various mesenchymal states, leading to significant differences in cell fates and functions[29–31]. It is likely that different forms of EndMT and EMT are controlled by transcription factors that exert distinct roles[30,31]. For example, SNAI1 and PRRX1 drive different modes of EMT during vertebrate development[30]. By analogy, our data suggest that TWIST1 regulates protective forms of EndMT in advanced atherosclerosis, and we propose that other unidentified factors contribute to the pathogenic forms of EndMT. Further research is needed to test this hypothesis and better understand the functional diversity of EndMT in atherosclerosis.

In this study, we delineated the role of TWIST1 in driving EndMT within atherosclerotic plaques, identifying its involvement in migration and proliferation pathways. TWIST1 achieves this by inducing the expression of multiple genes, with our analysis focusing on PELP1 and AEBP1. We found that PELP1 is required for both migration and proliferation, while AEBP1 is specifically required for proliferation. The involvement of PELP1 in EndMT is consistent with its known role in EMT and subsequent cell migration in breast cancer[32,33]. Our findings reveal a novel role for AEBP1 in EndMT, aligning with its established involvement in EMT[29] and epithelial cell proliferation[34]. AEBP1 acts as a central regulator of collagen, with mutations in the AEBP1 gene linked to defective collagen assembly in Ehlers-Danlos syndrome[22]. Additionally, AEBP1 has been implicated in regulating cell plasticity and activation in fibrosis[35,36], as well as upregulating collagen production[37,38]. We demonstrated that AEBP1 induces endothelial expression of COL4A1, a basement membrane component critical for vascular homeostasis[39–41] and cell proliferation[42,43]. Indeed, COL4A1 mutations are associated with coronary artery disease[44] and stroke[45]. Our findings suggest that COL4A1 promotes the proliferation of EndMT cells downstream of the TWIST1-AEBP1 pathway. Further research is needed to determine whether COL4A1 contributes to plaque stability through this mechanism. Overall, TWIST1 regulates EndMT via PELP1 and AEBP1-COL4A1 signaling, promoting enhanced migration and proliferation.

*Twist1*-driven EndMT and plaque stability were observed in male but not female mice, highlighting sex-specific differences in atherosclerosis mechanisms. In women, atherosclerosis often presents as diffuse involvement of the cardiovascular system, including microvascular disease, and is associated with a more stable plaque phenotype. In contrast, men typically develop focal disease with larger, more vulnerable plaques[46,47]. Notably, plaque rupture and erosion, key triggers of stroke, unstable angina, and myocardial infarction, exhibit sex-specific patterns, with plaque erosion most common in premenopausal women[46,47]. Our observation that atherosclerosis is unaltered in female *Twist1*[ECKO] mice is consistent with Shin et al. who demonstrated that EndMT is more prominent in atherosclerotic lesions of male *ApoE*[−/−] mice compared to females[48]. These findings highlight the need for further research into the sex-dependent role of EndMT in atherosclerosis.

In summary, we identify TWIST1 as a critical regulator of specific clusters of EndMT cells in advanced atherosclerosis. TWIST1-dependent EndMT enhances endothelial migration via PELP1 and activates an AEBP1-COL4A1 pathway to drive endothelial proliferation. Our findings challenge the prevailing view that EndMT solely destabilizes plaques, instead highlighting a TWIST1-driven program that promotes plaque stability to reduce rupture risk. Further research is necessary to unravel the molecular regulation of EndMT subtypes in atherosclerosis and assess the potential of TWIST1 as a therapeutic target for stabilizing vulnerable plaques.

## Methods
### Mice
Animal care and experimental procedures were carried out under licenses issued by the UK Home Office, and local ethical committee approval was obtained from the Animal Welfare and Ethical Review Body at the University of Sheffield. Mice were housed in open cages with wood bedding (12/12 h light/dark cycle) at 21 °C and 50% humidity. Mice had *ad libitum* access to food and water. All mice were on a C57BL/6 J background. *Twist1*[fl/fl] mice were obtained from Dr. Akiko Mammoto (Medical College of Wisconsin, USA). *Rosa26*[TdTomato] was kindly supplied by Professor Nicola Smart (University of Oxford, UK). *Cdh5*[CreERT2/+] and *ApoE*[−/−] colonies were obtained from Professor Sheila

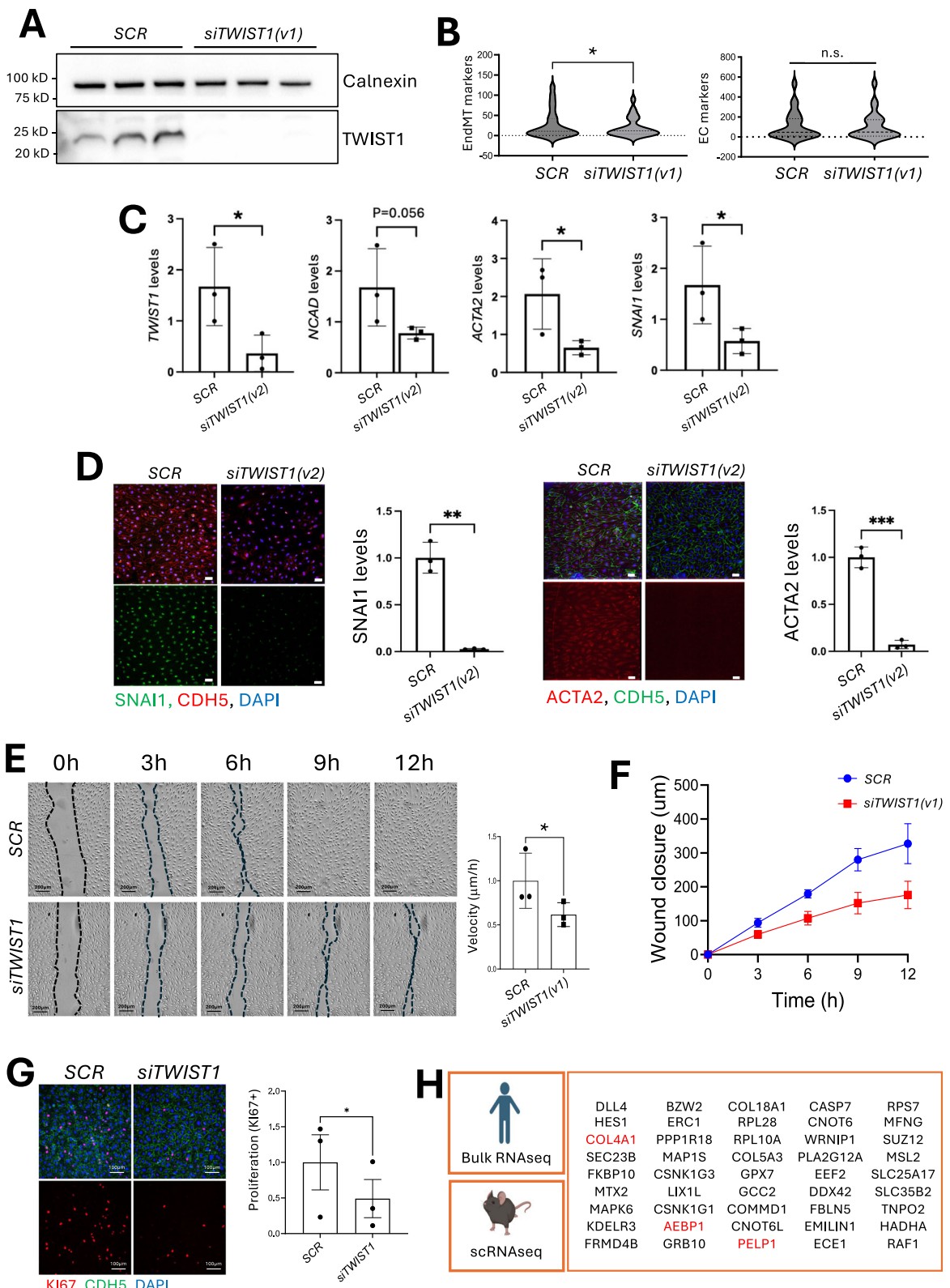

Francis (University of Sheffield, UK). Inducible deletion of *Twist1* was performed by crossing *Twist1^{fl/fl} Cdh5^{Cre-ERT2/+}* with *ApoE^{−/−}* mice to generate *Twist1^{ECKO}* mice *(Twist1^{fl/fl} Cdh5^{CreERT2/+} ApoE^{−/−})* and control mice *(Twist1^{fl/fl} Cdh5^{+/+} ApoE^{−/−})*. Fate mapping in combination with inducible deletion was performed by crossing *Twist1^{fl/fl} CDH5^{Cre-ERT2/+} ApoE^{−/−} Rosa26^{tdTomato/tdTomato}* with *ApoE^{−/−} Rosa26^{tdTomato/tdTomato}* mice to generate *Twist1^{ECKO} Rosa26 (Twist1^{fl/fl} Cdh5^{CreERT2/+} ApoE^{−/−} Rosa26^{TdTomato/TdTomato})* and control *Rosa26 (Twist1^{+/+} Cdh5^{CreERT2/+} ApoE^{−/−} Rosa26^{TdTomato/TdTomato})*

mice. Genotyping was performed using PCR primers described in Supplementary Table 1. Mice aged 8 weeks old were given a Western diet (TD.88137, Envigo) for 8 weeks. Experimental and control groups were treated by intraperitoneal administration of tamoxifen (Sigma-Aldrich) in corn oil for five consecutive days (2 mg per mouse per day), followed by a Western diet for 6 weeks. This led to activation of CRE in mice containing *Cdh5^{CreERT2}*. An additional cohort of *Twist1^{+/+} Cdh5^{+/+} ApoE^{−/−}* mice was given a Western diet (TD.88137, Envigo) for 8 weeks

**Fig. 3 | TWIST1 promotes EndMT, proliferation and migration of arterial EC.** HAECs from three independent donors were transfected with *TWIST1* siRNA (*siT-WIST1(v1) or siTWIST1(v2)*) or scrambled (SCR) siRNA. Transfected EC were exposed to DF for 72 h using the Ibidi system (**A**, **B**, **G**) or orbital shaker (**C**–**F**). **A** TWIST1 knockdown was confirmed by Western blotting (*n* = 3). **B** Violin plot of EMT-related gene expression (left) and EC-specific markers (right) from bulk RNAseq. **C** qRT-PCR quantitation of *TWIST1* and the EndMT markers *NCAD, ACTA2* and *SNAI1* (*n* = 3 donors). **D** Immunofluorescent staining of SNAI1 (left) and ACTA2 (right) co-stained with anti-CDH5 (EC) and DAPI (nuclear stain, blue) (*n* = 3 donors; scale bar = 50 μm). **E**, **F** Cell migration was assessed using a scratch wound assay in *SCR* vs *siTWIST1* HAEC monolayers after exposure to DF (orbital shaker). **E** Brightfield images show wound closure at different time points (scale bar = 200 μm). Average migration velocity over 12 h was quantified (distance migrated/12 h) (*n* = 3 donors). **F** The distance migrated from the initial wound (T0) was measured at multiple time points (*n* = 3 donors). **G** Ki67 immunofluorescence staining (red) was performed to quantify proliferation in *SCR* vs *siTWIST1*-treated HAEC under DF (Ibidi system). Merged images show DAPI (nuclear stain, blue) and VE-cadherin (green) (scale bar = 100 μm). Ki67-positive cells were quantified as a proportion of total nuclei, and values are shown as relative fold change compared to *SCR* (*n* = 3 donors). **H** RNA-seq analysis of HAECs was compared with scRNA-seq from murine plaque endothelium, identifying 45 common *Twist1*-regulated genes involved in EMT, EndMT, proliferation, and migration. Created in BioRender. Evans, P. (2026) https://BioRender.com/hdd2ran. Mean values are shown ± standard errors. Differences between means were analysed using a two-sided unpaired *t*-test (**C**, **D**), a two-sided ratio paired *t*-test (**E**), a one-sided ratio paired *t*-test (**G**) or a two-sided Wilcoxon test (**B**). * $P < 0.05$; ** $P < 0.01$; *** $P < 0.001$; n.s. = not significant. Source data are provided as a Source Data file.

prior to plaque analysis to provide baseline measurements. Mice were euthanised by pentobarbital intraperitoneal injection followed by exsanguination.

## Histological analysis of atherosclerotic plaques

Mice were euthanized and perfusion-fixed via the left ventricle with phosphate-buffered saline (PBS), followed by 4% paraformaldehyde (PFA). The brachiocephalic artery and aortic sinus were dissected, carefully cleared of adventitial tissue, and embedded in paraffin. Paraffin blocks were sectioned at 5 μm thickness. Sections were dewaxed in xylene and rehydrated through graded alcohols using standard protocols prior to histological analysis. Elastic lamina morphology and plaque quantification were assessed using Miller's elastin stain. The internal elastic lamina was delineated, and the areas of the intima and plaque were quantified using ImageJ. Plaque size was expressed as the percentage of the intimal area occupied by plaque. Hematoxylin and eosin (H&E) staining was used to measure the area of necrotic, acellular material within the plaque, expressed as a percentage of total plaque area. The proportion of plaque area within the intima was also independently verified using H&E staining. Collagen content was assessed using Picrosirius Red staining and expressed as the percentage of plaque area occupied by collagen. The proportions of vascular smooth muscle cells (VSMCs; anti-ACTA2) and macrophages (anti-MAC3; antibodies listed in Supplementary Table 2) were determined by immunohistochemistry and expressed as percentages of plaque area. For antigen retrieval, sections were boiled in citrate buffer for 10 min. Breaks in elastin fibers within the intima adjacent to plaques were manually counted. All sections were imaged using a Leica Thunder microscope, Nikon ECLIPSE microscope or NanoZoomer S60, and quantitative analyses were performed using ImageJ.

## Immunostaining of atherosclerotic plaques

Brachiocephalic arteries were placed in 30% sucrose for 72 h before embedding in optical cutting temperature compound and snap-freezing. Blocks were sectioned at 10 μm intervals. Frozen sections were washed with PBS and permeabilised with 0.5% Triton X-100 for 20 min. Sections were blocked with 1% bovine serum albumin (BSA), 10% horse serum and stained with primary antibodies (anti-VWF, anti-ACTA2, anti-COL4A1, anti-AEBP1 or anti-PELP1) (see Supplementary Table 2) overnight at 4 °C. Sections were then washed with PBS, incubated with Alexa Fluor 488-conjugated secondary antibodies for 1 h at room temperature, washed again with PBS, incubated with DAPI and mounted with ProLong Gold mounting reagent. Frozen sections were imaged using a Zeiss LSM 980 Airyscan confocal microscope and analysed using ImageJ and Qupath.

## Plasma lipid measurements

Blood samples collected by terminal cardiac puncture were subjected to centrifugation to obtain plasma. Plasma cholesterol and triglyceride levels were measured using a COBAS analyser (total plasma cholesterol, non-high-density lipoprotein cholesterol, high-density lipoprotein cholesterol, low-density lipoprotein cholesterol and triglycerides).

## Mouse aorta digestion and FACS for scRNA-seq

The left ventricle of the murine heart was perfused with heparin sodium (20 U/ml) in PBS. Aortas from male *Twist1*[ECKO] (*Twist1*[fl/fl] *Cdh5*[CreERT2/+] *ApoE*[−/−]) (*n* = 5) and control (*Twist1*[fl/fl] *Cdh5*[+/+] *ApoE*[−/−]) (*n* = 4) mice after 14 weeks of Western diet were dissected from the aortic root to the iliac artery, removing fat and connective tissue. Aortas were then transferred into PBS and incubated in collagenase type I (450 U/ml), collagenase type XI (125 U/ml), hyaluronidase type 1-s (60 U/ml), deoxyribonuclease I (60 U/ml), and elastase (0.5 mg/ml) in PBS for 10 min at 37 °C, to enable the removal of the adventitial layer. Aortas were then cut into 2 mm pieces and further incubated in the enzymatic solution for 1 h and 15 min at 37 °C to generate a single-cell suspension. When the aortas were digested, the single-cell suspensions were filtered through a 40-μm cell strainer, washed, and resuspended in 1% BSA solution in PBS. Single-cell suspensions were incubated for 5 to 10 min with TruStain FcX (anti-mouse CD16/32) antibody to block unspecific binding of immunoglobulin to Fc receptors, and they were subsequently stained with allophycocyanin (APC)-conjugated anti-CD45 and Alexa Fluor (AF) 488–conjugated anti-CD31, washed and diluted in 1% BSA in PBS. To exclude dead cells, samples were stained with TOPRO-3. CD31[+], CD45[−], and TOPRO-3[−] cells were sorted into 384-well plates containing reverse transcription primers, deoxyribonucleotide triphosphates, External RNA Controls Consortium (ERCC) RNA Spike-Ins, and mineral oil using a BDFACSMelody cell sorter (BD Biosciences). Plates containing capture cells were snap-frozen and stored at −80 °C until sequencing was performed.

## Single-cell RNA sequencing

scRNA-seq libraries were generated using the SORT-seq approach as described previously[49]. After lysing single cells at 65 °C for 5 min, reverse transcriptase and second-strand mixers were added to the wells using the Nanodrop II liquid handling platform (GC Biotech). The mRNA of each individual cell was reverse-transcribed, and double-stranded complementary DNAs were pooled, and in vitro transcription was carried out for linear amplification of RNA. The Illumina sequencing libraries were generated using TruSeq small RNA primers (Illumina), and the Illumina NextSeq (carried out commercially by Single Cell Discoveries, Utrecht, The Netherlands) was used to sequence paired-end at 75–base pair read length the DNA libraries. The RNA yield of the amplified RNA and the quality and concentration of the final cDNA libraries were measured using Bioanalyzer (Agilent).

## Bioinformatics

Reads from Illumina sequencing were aligned to the GRCm38 mouse genome using STAR. Single-cell transcriptomes generated using SORT-

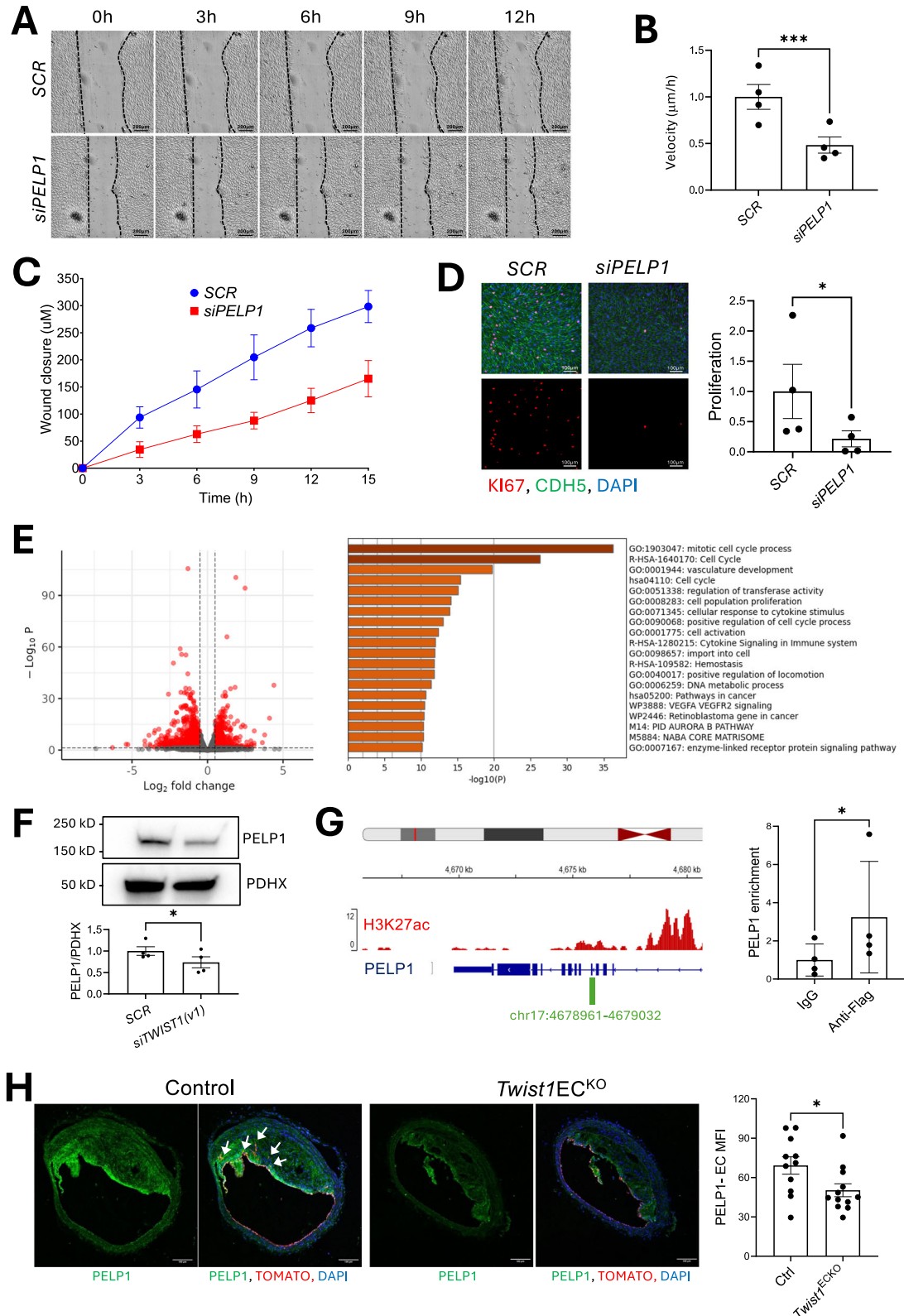

seq were quality-filtered, excluding cells with less than 1800 transcripts per cell and more than 10% mitochondrial content from further analysis (2139 cells out of 3043 total sorted cells were analysed). After filtering count data and selecting high-quality single cells, the data were normalized, and the top 2000 highly variable genes in our dataset were selected and used for principal component analysis. The first 20 principal components were retained based on their

contribution to overall variability and used for downstream analysis, to perform dimensional reduction and clustering. Dimensional reduction and clustering of our scRNA-seq dataset were performed using the R package Seurat (v3.1), and a clustering resolution of 0.8 was used. Cell cycle regression was performed by calculating an aggregated score for S and G2M genes per cell to mitigate the effects of cell cycle heterogeneity. Differential expression analysis was performed between

**Fig. 4 | TWIST1 promotes migration and proliferation via PELP1. A–C** Cell migration was assessed using a scratch wound assay in *SCR* vs *PELP1*-silenced (*siPELP1*) HAEC monolayers exposed to DF (orbital shaker). HAEC were from four independent donors. **A** Brightfield images show wound closure at different time points for a single representative donor (scale bar = 200 μm). **B** Average migration velocity over 12 h was quantified (distance migrated/12 h) (*n* = 4 donors). **C** The distance migrated from the initial wound (T0) was measured at multiple time points (*n* = 4 donors). **D** *SCR*- and *siPELP1*-treated HAECs were exposed to DF for 72 h (orbital shaker) and stained using antibodies against Ki67 (red) to assess proliferation. Merged images show DAPI (nuclear stain, blue) and VE-cadherin (green) (scale bar = 100 μm). Ki67-positive cells were quantified as a proportion of total nuclei, and values are shown as relative fold change compared to *SCR* (*n* = 4 donors). **E** RNA-seq analysis comparing control (*SCR*) and *PELP1*-silenced HAECs exposed to DF using the orbital shaker (*n* = 4 donors). RNA-seq data were analyzed using the DESeq2 Bioconductor package, incorporating donor as a covariate to account for donor-specific effects observed in principal component analysis. Multiple testing correction was performed using the Benjamini-Hochberg procedure. On the left, volcano plot showing differentially expressed genes (Padj < 0.05, fold change > 1.5, highlighted in red). On the right, pathway enrichment analysis (Metascape) identifies key pathways enriched by *PELP1* siRNA. **F** Western blot analysis of PELP1 expression in *SCR* vs si*TWIST1(v1)*-silenced HAECs exposed to DF (Ibidi system), normalized to PDHX (*n* = 4 donors). **G** HAECs were infected with lentivirus expressing TWIST1-FLAG and exposed to DF using an orbital shaker for 72 h. TWIST1 binding at the *PELP1* loci was assessed using an anti-FLAG antibody, with IgG as a control. On the left, a schematic representation of the *PELP1* gene locus, with TWIST1 binding sites highlighted in green. On the right, ChIP-qPCR fold enrichment relative to the IgG control (*n* = 4 donors). **H** Brachiocephalic artery atherosclerotic plaques after 14 weeks of Western diet. Frozen sections of brachiocephalic arteries were stained using antibodies against PELP1 (green) in *Twist1*[ECKO] (*n* = 12) and control (*n* = 11) mice. Rosa26TdTomato+ cells are shown in red, and nuclei are counterstained with DAPI (blue). Arrows mark Rosa26TdTomato-PELP1 double-positive cells in the plaque. Representative images are shown (Scale bar = 100 μm). Mean values are shown ± standard errors. Differences between means were analysed using a two-tailed ratio paired *t*-test (**B, D, G**), a one-tailed ratio paired *t*-test (**F**) or a two-sided unpaired *t*-test (**H**). * *P* < 0.05; *** *P* < 0.001. Source data are provided as a Source Data file.

---

clusters, and marker genes for each cluster were determined with the Wilcoxon rank sum test with *P* < 0.001 using Seurat and the single-cell analysis platform BBrowserX. Differential expression analysis, heatmaps of gene expression embedded on hierarchical clustering, t-distributed stochastic neighbor embedding (t-SNE) representations showing the expression of defined genesets/GO pathways, and expression of single transcripts on the t-SNE embedding were performed using the software BBrowserX. Violin plot representations of the expression of defined GO pathways shows the AUCellScore, which uses the "Area Under the Curve" to calculate whether a critical subset of selected GO pathways is enriched within the expressed genes of each cell.

### Cell culture and flow experiments
HAECs were obtained from PromoCell and cultured in EGM2 growth medium (#C-22011) according to the manufacturer's guidelines. HAECs from four different donors (three male, one female), non-pooled, were used for all experiments between passages 3 and 7. Cells were exposed to flow conditions using either the Ibidi flow apparatus or an orbital shaker. For Ibidi flow, cells were seeded on gelatin or collagen IV-coated Ibidi μ-Slides I$^{0.4}$ (Luer ibiTreat, Ibidi) and used when cells reached full confluence. UF consisted of 13 dynes/cm² unidirectional shear stress. To generate DF, HAECs were subjected to a cyclic regimen of 2 h of oscillatory flow (± 4 dynes/cm², 0.5 Hz), followed by 10 min of unidirectional flow (+ 4 dynes/cm²) to promote nutrient redistribution. The slides and pump system were maintained in a cell culture incubator at 37 °C during the experiment. For orbital flow, HAECs were grown to confluence in gelatin-coated 6-well plates and exposed to flow for 72 h on an orbital shaking platform (PSU-10i; Grant Instruments) placed inside a cell culture incubator. The shaker had a 10 mm orbit radius and was set at 210 revolutions per minute (RPM), generating DF (approximately 5 dynes/cm² multidirectional shear stress) at the well center and uniform flow with higher shear stress (approximately 10 dynes/cm²) and stable directionality at the periphery.

### Gene silencing
HAECs were transfected with siRNA sequences targeting *TWIST1* (L-006434-00, Dharmacon; *siTWIST1(v1)*) or Silencer Select s14523, Ambion; *siTWIST1(v2)*), *AEBP1* (s1145, Ambion), *COL4A1* (s3289, Ambion), *DLL4* (s29215, Ambion) *FKBP10* (s34173, Ambion), *KDELR3* (s21690, Ambion), *MTX2* (s20935, Ambion), *RPS7* (s12290, Ambion), (s34173, Ambion), *PELP1* (s25746, Ambion), SEC23B (s20536, Ambion), *USP14* (s17358, Ambion) using the Lipofectamine RNAiMAX transfection system (13778-150, Invitrogen), as per the manufacturer's protocol. Scrambled siRNA (D-001810-01-50, Dharmacon) was used as a control. Experiments were conducted with 25 nM siRNA, and cells were incubated in transfection media for 6 h before media replacement and exposure to flow.

### Quantitative real-time PCR (qRT-PCR)
Total RNA was extracted from HAECs using the RNeasy Mini Kit (Qiagen, 74104) and reverse transcribed into cDNA using the iScript cDNA Synthesis Kit (Bio-Rad, 1708891). Gene expression was measured by qRT-PCR with specific primers (Supplementary Table 2) and SsoAdvanced Universal SYBR Green Supermix (Bio-Rad, 172-5271). Expression levels were normalized to the housekeeping gene HPRT, and fold changes were calculated using the ΔΔCt method.

### Bulk RNA sequencing of cultured EC
Total RNA from HAECs was extracted and quality-checked using a Bioanalyzer (Agilent). High-quality RNA was used for RNA-seq library preparation and sequenced on an Illumina platform, generating 40 million reads per sample. Library preparation and sequencing were performed by Novogene. Fastq files were processed using the nf.core rna-seq pipeline (version 3.6.0) (https://nf-co.re/rnaseq/3.6), incorporating salmon pseudo-alignment and quantification. The DESeq2 Bioconductor package was used for differential expression, with a donor effect added to the model to compensate for donor-specific effects in expression observed with a PCA plot. Functional enrichment analysis was conducted using Metacore/Metascape on protein-coding genes with *P* < 0.05.

### Western Blot
Flow-exposed cells were washed twice with PBS and lysed in Laemli/RIPA buffer containing β-mercaptoethanol. Lysates were separated on 4–15% Mini-PROTEAN® TGX™ Precast Protein Gels (Biorad #4561083) for 90 min at 100 V, followed by transfer to PVDF membranes (Bio-Rad, #1660827 EDU) at 230 mA for 60 min. Membranes were blocked for 1 h with 10% milk in TBS-T, then stained overnight with primary antibodies (Supplementary Table 3) in 5% milk. After washing, secondary antibodies (Goat Anti-Mouse or Goat Anti-Rabbit, Dako 0447 and P0448) were applied at 1:3000 in 5% milk for 1 h. Blots were visualized using ECL Select (Amersham, #RPN2235) and the Chemidoc XRS system (Bio-Rad).

### Immunofluorescence of cultured EC
HAECs were washed twice with PBS, fixed for 10 min in 4% paraformaldehyde, permeabilized with 0.1% Triton X-100, and blocked with 20% goat serum. Cells were incubated with primary antibodies (Supplementary Table 3) in 5% serum overnight at 4 °C. After washing, secondary antibodies conjugated with Alexa Fluor were applied, and

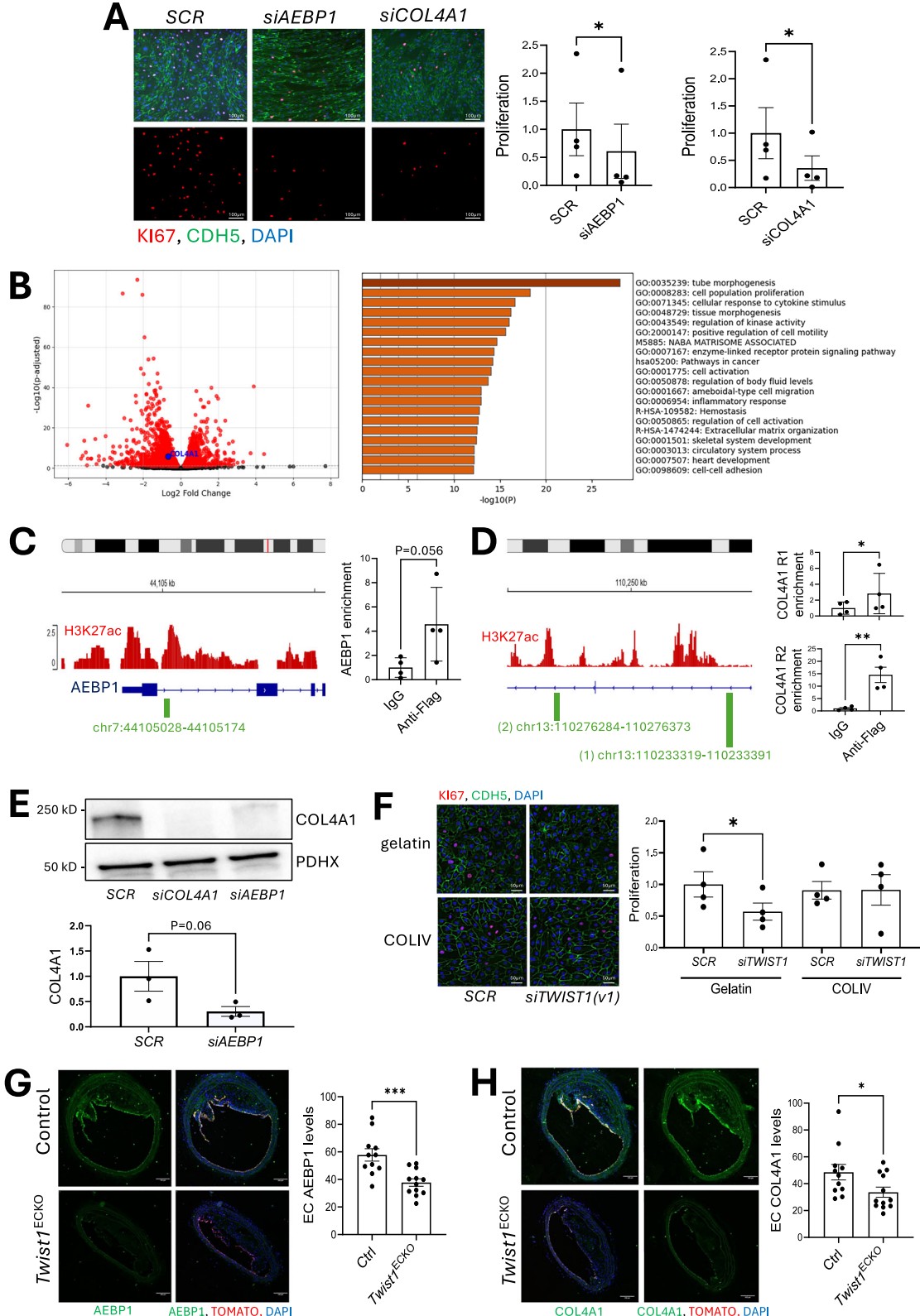

images were captured using a Leica Thunder, LSM980 Confocal, or ZEISS CD7 microscope.

**Migration assay**

HAECs grown on gelatin-coated 6-well plates and silenced for TWIST1, PELP1, or scrambled siRNA were exposed to 72 h of orbital flow-induced DF. Cells were scratched using a pipette tip, washed, and fresh media was added. Wound closure was monitored using live imaging on the ZEISS CD7 microscope at various times up to 24 h post-incubation. Migration rates were quantified by measuring the distance moved by cells from the scratch edge. The velocity was calculated as the distance/hour after 12 h assay.

**Fig. 5 | TWIST1 promotes proliferation of arterial EC via AEBP1-COL4A1 signaling. A–F** Cultured HAEC were analysed from multiple independent donors. **A** Control (*SCR*), *AEBP1*-silenced (*siAEBP1*), and *COL4A1*-silenced (*siCOL4A1*) HAECs were exposed to DF (orbital shaker) for 72 h, and Ki67 immunofluorescence staining (red) was performed to assess proliferation. Merged images show DAPI (blue) and CDH5 (green) (scale bar = 100 μm). Ki67-positive cells were quantified as a proportion of total nuclei, and values are shown as relative fold change compared to *SCR* (n = 4 donors). **B** RNA-seq analysis comparing *SCR* and *siAEBP1* HAECs after 72 h under DF (orbital shaker). RNA-seq data were analyzed using the DESeq2 Bioconductor package, incorporating donor as a covariate to account for donor-specific effects observed in principal component analysis. Multiple testing correction was performed using the Benjamini-Hochberg procedure. On the left, volcano plot of differentially expressed genes (Padj < 0.05, fold change > 1.5, highlighted in red). Among these, *COL4A1* is significantly regulated by *AEBP1* and is indicated. On the right, pathway enrichment analysis (Metascape) identifying key pathways enriched by *AEBP1* siRNA. **C, D** HAECs were infected with lentivirus expressing TWIST1-FLAG and exposed to DF using an orbital shaker for 72 h. TWIST1 binding at the *AEBP1* and *COL4A1* loci was assessed using an anti-FLAG antibody, with IgG as a control. **C** On the left, a schematic representation of the

*AEBP1* gene locus, with TWIST1 binding sites highlighted in green. On the right, ChIP-qPCR fold enrichment relative to the IgG control (n = 4 donors). **D** On the left, a schematic representation of the *COL4A1* gene locus, with TWIST1 binding sites highlighted in green. On the right, ChIP-qPCR fold enrichment relative to the IgG control (n = 4 donors). **E** Western blot analysis of COL4A1 levels in control (*SCR*), *AEBP1*-silenced (*siAEBP1*) and *COL4A1*-silenced (*siCOL4A1*) HAECs (n = 3 donors). **F** Proliferation was quantified in si*TWIST1(v1)*-silenced cells grown on gelatin or a collagen IV matrix by Ki67 immunofluorescence staining (red). Merged images show DAPI (blue) and CDH5 (green) (n = 4 donors). Scale bar = 50 μm.

**G, H** Brachiocephalic artery atherosclerotic plaques after 14 weeks of Western diet. Frozen sections of brachiocephalic arteries were stained using antibodies against **G** AEBP1 (green) and **H** COL4A1 (green) in *Twist1*^ECKO^ (n = 12) and control (n = 11) mice. Rosa26TdTomato⁺ cells are shown in red, and nuclei are counterstained with DAPI (blue). Representative images are shown (Scale bar = 100 μm). Mean values are shown ± standard errors. Differences between means were analysed using (**A, C, D**) a two-sided ratio paired *t*-test, (**E**) a one-sided ratio paired *t*-test, (**F**) a one-way anova or (**G, H**) a two-sided unpaired *t*-test. * *P* < 0.05; *** *P* < 0.001. Source data are provided as a Source Data file.

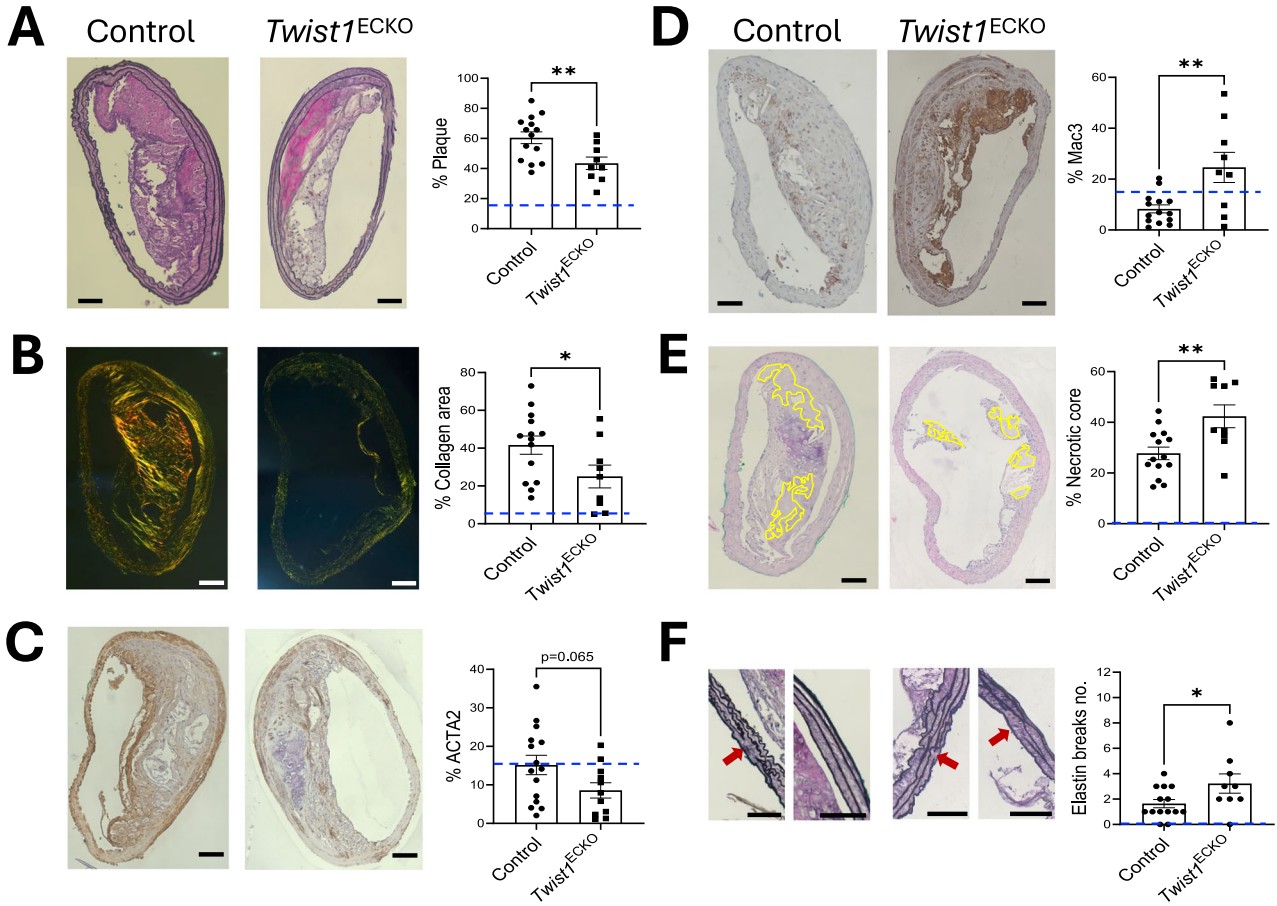

**Fig. 6 | *Twist1* promotes brachiocephalic plaque growth and features of stability in vivo.** Male *Twist1*^ECKO^ (*Twist1*^fl/fl^ *Cdh5*^CreERT2/+^ *ApoE*^−/−^) and control mice (*Twist1*^fl/fl^ *Cdh5*^+/+^ *ApoE*^−/−^) aged 8 weeks were fed a Western diet for 8 weeks to induce atherosclerotic lesions. Tamoxifen was then administered to both groups for 5 consecutive days to induce *Twist1* deletion in ECs of experimental mice. Mice were then fed a Western diet for an additional 6 weeks (totaling 14 weeks of Western diet). Paraffin-embedded sections of brachiocephalic arteries were stained with **A** Miller's elastin stain to quantify plaque size, **B** Picrosirius Red (visualized under polarized light) to quantify collagen content, **C** antibodies against ACTA2 to quantify vSMCs content, **D** antibodies against MAC3 to quantify macrophage

content, and **E** Hematoxylin and Eosin (H&E) to quantify necrotic core (highlighted in yellow) content in *Twist1*^ECKO^ (n = 9) and control (n = 14) mice. **F** Magnified view of specific regions shown in (**A**) and quantification of elastin breaks number in brachiocephalic arteries from *Twist1*^ECKO^ (n = 9) and control (n = 14) mice. The red arrows indicate elastin breaks. Representative images are shown (Scale bar = 100 μm). Mean values are shown ± standard errors. Blue dashed line represents a baseline value measured in a cohort of *Twist1*^+/+^ *Cdh5*^+/+^ *ApoE*^−/−^ mice exposed to a Western diet for 8 weeks (see Fig. S20). Differences between means were analysed using a two-sided unpaired *t*-test. * *P* < 0.05; ** *P* < 0.01.

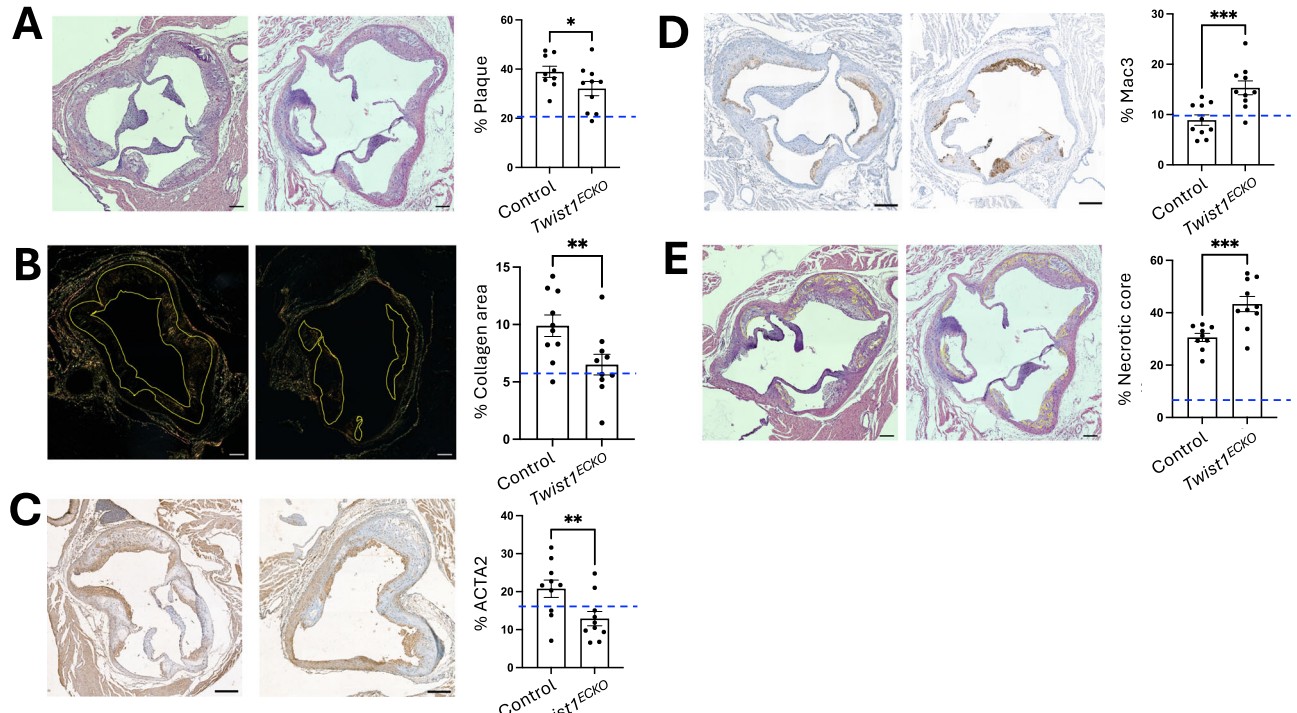

**Fig. 7 | *Twist1* promotes aortic root plaque growth and features of stability in vivo.** Male *Twist1*ECKO (*Twist1*fl/fl *Cdh5*CreERT2/+ *ApoE*−/−) and control mice (*Twist1*fl/fl *Cdh5*+/+ *ApoE*−/−) aged 8 weeks were fed a Western diet for 8 weeks to induce atherosclerotic lesions. Tamoxifen was then administered to both groups for 5 consecutive days to induce *Twist1* deletion in ECs of experimental mice. Mice were then fed a Western diet for an additional 6 weeks (totaling 14 weeks of Western diet). Paraffin-embedded sections of aortic roots were stained with **A** Hematoxylin and Eosin (H&E) to quantify the plaque size in *Twist1*ECKO (*n* = 10) and control (*n* = 9) mice, **B** Picrosirius Red (visualized under polarized light) to quantify collagen content in *Twist1*ECKO (*n* = 10) and control (*n* = 10) mice, **C** antibodies against ACTA2 to quantify vSMCs content in *Twist1*ECKO (*n* = 10) and control (*n* = 10) mice, **D** antibodies against MAC3 to quantify macrophage content in *Twist1*ECKO (*n* = 10) and control (*n* = 10) mice, **E** Hematoxylin and Eosin (H&E) to quantify necrotic core (highlighted in yellow) content in *Twist1*ECKO (*n* = 10) and control (*n* = 9) mice. Representative images are shown (Scale bar = 100 μm). Mean values are shown ± standard errors. Blue dashed line represents a baseline value measured in a cohort of *Twist1*+/+ *Cdh5*+/+ *ApoE*−/− mice exposed to a Western diet for 8 weeks (see Fig. S21). Differences between means were analysed using a one-sided unpaired t-test. * *P* < 0.05; ** *P* < 0.01; *** P < 0.001. Source data are provided as a Source Data file.

## Chromatin immunoprecipitation (ChIP)

ChIP assays were performed using the Diagenode iDeal ChIP qPCR kit, as per the manufacturer's instructions. HAECs overexpressing TWIST1-Flag were subjected to 72 h of orbital flow before cross-linking with 1% formaldehyde. Cells exposed to DF were scratched and lysed to isolate nuclei. chromatin was sheared via sonication 30 s ON/ 30 s OFF on a bioruptor plus sonicator (Diagenode). Immunoprecipitation was performed using protein A magnetic beads and 10 μg of ChIP-grade anti-Flag or rabbit IgG antibody for 3 h, followed by overnight incubation with sheared chromatin. Immunoprecipitated DNA was quantified using qPCR with primers listed in (Supplementary Table 2). DNA enrichment over input was calculated, and the data were normalized to IgG.

## Lentiviral overexpression

Vectors for lentiviral expression of FLAG-tagged versions of TWIST1, PELP1, AEBP1 and COL4A1 were obtained commercially (pGenLenti; Addgene). HEK 293 T cells were plated on Day 1 to reach 80–90% confluency by the following day. On Day 2, cells were transfected with a plasmid mix containing 4,2 pm of Pax2, VSVG, and pGenLenti-cDNA vector using PEI (polyethylenimine). The plasmid-PEI complex was incubated for 20 min before being added to the cells. After 6 h, the media was replaced with MEM (+10% FCS, glutamine, Pen/Strep). On Day 3, the media was changed to EGM2. Viral supernatants were collected on Days 4 and 5, centrifuged at 500 g, filtered through a 0.45 μm filter, and transferred to HAECs at a 1:1 ratio.

## Human carotid plaque analysis

Human plaque bulk RNA-seq and immunostaining of TWIST1 were performed on carotid plaques obtained from the CPIP biobank (Region Skåne, Malmö, Sweden). The study cohort consisted of patients undergoing carotid endarterectomy, with a median age of 70 years (interquartile range, 64–75) and 64% female participants. All patients have given written informed consent, and the study follows the declaration of Helsinki. The study protocol has been approved by the local ethical committee in Lund and the Swedish ethical committee (472/2005, 2014/904, 60/2008, 2012/209). The indications for surgery were: 1) asymptomatic carotid stenosis with a stenosis degree >80% or 2) cerebrovascular symptoms (ischemic stroke, transient ischemic attack, or amaurosis fugax within one month prior to surgery) and a carotid plaque with a stenosis degree >70%, as previously described[50]. For follow-up analysis, information regarding postoperative cardiovascular events (myocardial infarction, unstable angina, stroke (ipsilateral and contralateral events), transient ischemic attack, amaurosis fugax, vascular interventions (including carotid endarterectomy/stenting, coronary artery bypass grafting/percutaneous coronary artery intervention) and CV death was acquired from the Swedish Cause of Death and National inpatient Health Registers. The participants were followed until events or the end of follow-up by the 31st December 2015.

All plaques were immediately snap frozen in liquid nitrogen upon surgical removal. To assess TWIST1 mRNA levels, one cross-sectional fragment of 1 mm was taken from the most stenotic part of all plaques.

RNA was isolated using Trizol and sequenced using both Illumina HiSeq2000 and NextSeq 500 platforms, as described previously[51].

In short, transcript-level quantification was conducted using Salmon based on transcriptome release 27 of GENCODE in mapping-based mode. Gene counts were summarized using tximport and were normalized between samples using a trimmed mean of M-values (TMM) by edgeR[52], giving gene expressions as log$_2$-transformed counts per million (CPM) after voom transformation. Batch effects of sequencing platforms were adjusted by an empirical Bayes method[53]. Formalin-fixed paraffin-embedded sections 4 µm thick were stained using anti-TWIST1 antibodies (Abcam, ab175430; 5 ug/mL) and positive immunoreactivity was detected using a MACH3 probe coupled with horseradish peroxidase polymer (Biocare Medical, Pacheco, CA, USA). Isotype controls were used to confirm the specificity of primary antibody binding and to exclude non-specific background staining. The slides were then counterstained with Mayer's Hematoxylin (HistoLab, Göteborg, Sweden). Whole-slide images were scanned and imported into QuPath (v0.6.0).

### Statistics and reproducibility

Sample sizes were selected based on power calculations using preliminary data. No data were excluded from the analyses. Allocation to experimental groups was determined by the random inheritance of the Cre allele, which occurs at approximately 50% frequency. Mice carrying the Cre allele were designated as *Twist1$^{ECKO}$*, whereas mice lacking the Cre allele served as controls. The Investigators were blinded to the identity of the samples from *Twist1$^{ECKO}$* and control mice during experiments and outcome assessment. Data are presented as mean values ± standard error of the mean. GraphPad Prism analysis software was used to carry out statistical analyses. Significance is shown as follows: *$p < 0.05$; **$p < 0.01$; ***$p < 0.001$, ****$p < 0.0001$. Specific tests used for each experiment are explained in the figure legends.

### Reporting summary

Further information on research design is available in the Nature Portfolio Reporting Summary linked to this article.

## Data availability

The fully annotated scRNA-seq and bulk RNA-seq generated in this study have been deposited in the Gene Expression Omnibus database under accession code GSE293412. The quantitative data generated from the analysis of mice and human cells are provided in the Source Data file. The human data is protected due to privacy laws and would be shared in group form upon request from a qualified academic investigator for the sole purpose of replicating the procedures and results presented in the article and providing that the data transfer is in agreement with European Union legislation on the general data protection regulation and decisions by the ethical review board of Sweden, the Region Skåne and the Lund University. Professor Isabel Goncalves (Isabel.Goncalves@med.lu.se) may be contacted for access to human data. Data regarding living subjects cannot be publicly available due to the sensitive nature of the data regulated by GDPR. Source data are provided with this paper.

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

## Acknowledgments

This research was funded through a British Heart Foundation Program Award (RG/19/10/34506). This study was also supported by the National Natural Science Foundation of China-Royal Society Joint Program (Grant Nos. 12411530127). The work was also supported by the Swedish Society for Medical Research [AE, CG-22-0254-H-02], the Swedish Research Council [AE, 2024-02761; IG, 2019-01260, 2023-02368] Swedish Heart and Lung Foundation [AE, 20220044 and 20220284; I.G, 20200403], the Swedish Stroke Association [J.S, S-993166], Hjelt Diabetes Foundation [J.S], SUS foundations and funds [A.E; I.G], Leducq Foundation [IG], Lund University Diabetes Center (Swedish Research Council - Strategic Research Area Exodiab Dnr 2009-1039 and the Swedish Foundation for Strategic Research Dnr IRC15-0067). MS was partially supported by NIH grant HL167014. The Knut and Alice Wallenberg Foundation, the Medical Faculty at Lund University and Region Skåne are acknowledged for generous financial support [AE]. We acknowledge Lena Sundius and Fiona Wright for their excellent histology support.

## Author contributions

B.T.A., M.D. (Diagbouga), A.D., S.T., A.E., E.C., J.S., C.G., M.D. (Dunning), X.L., and I.G. contributed to the design of experiments, acquisition and analysis of data and the preparation of the paper. J.K., M.S., and H.F.J. contributed to the design of experiments, analysis of data and preparation of the paper. S.E.F., A.M. and J.S.C. revised the manuscript for intellectual input. S.X. contributed to the design of experiments, analysis of data and preparation of the paper. P.C.E. conceived the study, designed the experiments, contributed to the acquisition of data, analysed the data and wrote the paper. All authors reviewed the paper and provided intellectual content.

## Competing interests

The authors declare no competing interests.

## Additional information

[1]Centre for Microvascular Research, William Harvey Research Institute, Barts and the London School of Medicine and Dentistry, Queen Mary University of London, London, UK. [2]Cardiovascular Research Translational Studies, Clinical Sciences Malmö, Lund University, Lund, Sweden. [3]Department of Cardiology, Skåne University Hospital, Malmö, Sweden. [4]Wallenberg Centre for Molecular Medicine, Lund University, Lund, Sweden. [5]Department of Biomedicine, Aarhus University, Høegh-Guldbergs Gade 10, 8000 Aarhus, Denmark. [6]Steno Diabetes Center Aarhus, Aarhus University Hospital, Aarhus, Denmark. [7]School of Medicine and Population Health, INSIGNEO Institute, University of Sheffield, Sheffield, UK. [8]Department of Pharmacy, Affiliated Hospital of Southwest Medical University, Luzhou, Sichuan 646000, China. [9]School of Pharmacy, Southwest Medical University, LuZhou, Sichuan 646000, China. [10]Department of Pediatrics, Medical College of Wisconsin, Milwaukee, WI 53226, USA. [11]Department of Cell Biology, Neurobiology and Anatomy, Medical College of Wisconsin, Milwaukee, WI 53226, USA. [12]Yale Cardiovascular Research Center, Yale University School of Medicine, New Haven, CT, USA. [13]Section of Cardiorespiratory Medicine, Department of Medicine, University of Cambridge, Victor Phillip Dahdaleh Heart and Lung Research Institute, Papworth Road, Cambridge CB2 0BB, UK. [14]Department of Endocrinology, the First Affiliated Hospital of USTC, Division of Life Sciences and Medicine, University of Science and Technology of China, Hefei 230001, China. [15]These authors contributed equally: Blanca Tardajos Ayllon, Mannekomba Diagbouga. ✉e-mail: paul.evans@qmul.ac.uk

