## [Transparent Peer Review file · Nature Communications]

TWIST1 drives endothelial-to-mesenchymal-transition to stabilize atherosclerotic plaques

Corresponding Author: Professor Paul Evans

Version 0:

Reviewer comments:

Reviewer #1

(Remarks to the Author)

The role of EndMT in atherosclerosis is important, but the mechanisms are still unclear. Previously, Dr. Evans and colleagues showed that a constitutional EC-targeted TWIST1 knockout reduces atherosclerotic plaque development. The current study further reports that TWIST1-driven EndMT plays an unexpected, plaque-stabilizing role in advanced atherosclerosis. Mechanistic studies show that TWIST1-dependent EndMT enhances endothelial migration via PELP1 and activates an AEBP1-COL4A1 pathway to drive endothelial proliferation. These results challenge the prevailing view that EndMT destabilizes plaques. Instead, it suggests a novel concept that TWIST1-driven EndMT promotes plaque stability, reducing rupture risk. Overall, the results support the conclusions. However, there are some concerns.

Specific Concerns:

Figure 6: For this study, the mice were treated with Tamoxifen for 5 days to induce TWIST1 ECKO at an 8-week time-point, followed by an additional 6-week Western diet. Were Control groups also treated with Tamoxifen to account for its potential atheroprotective complications? Did the authors also examine other atheroprone regions, including the aortic sinus and aortic arch? If so, were the results consistent in those regions as well?

Figure 7: TWIST1 protein expression in human carotid artery sections (n=3) is too small to support the authors' conclusion. The n number should be significantly increased to test the hypothesis.

Minor:

The sources of mice used to generate Twist1 ECKO ApoE^{-/-} and Twist1 ECKO-ApoE on an EC-fate-mapped background should be clarified.

Reviewer #2

(Remarks to the Author)

During endothelial-to-mesenchymal transition, ECs lose markers like VE-cadherin and acquire mesenchymal markers such as ACTA2. TWIST1 drives EndMT and promotes lipid accumulation in early atherosclerosis. Unexpectedly, the authors find that TWIST1 has a stabilizing effect on atherosclerotic plaques. An interesting relationship is demonstrated between clusters regarding level of Twist1 promotion and distribution of DF and UF markers, although the relationship with EndMT enrichment clusters is not as striking. The authors explore RNA expression in ECs and identify a set of differently expressed genes from which 10 candidates are prioritized for further assessment based on their TWIST1-dependent regulation. Among those, PELP1, AEBP1 and COL4A1 are singled out as the genes that regulated both proliferation and migration of ECs. Finally, the impact of EC Twist1 KO on atherogenesis is tested in male and female mice, revealing important sex differences. Interestingly, Twist1 ECKO leads to smaller plaques that feature a less stable phenotype, in accord with human data showing that reduced TWIST1 mRNA expression is associated with a greater risk of future cardiovascular events in patients. Overall, the study design is sound and the results convincing.

Comments:

Figure 2B show extensive Cdh5⁺ cells throughout the atherosclerotic lesions, presumably because of Cre recombinase reporter gene expression in haematopoietic cells. Is this representative? If so, how then are these cells accounted for in the

quantification of EndMT in Figure 2C? Also, how Twist1 in these cells may contribute to lesion formations should be considered.

The authors stipulate that effects of Twist1 on migration and proliferation primarily occur in DF cells but effects of Twist1KO on the function of ECs under UF should also be tested to confirm.

What approach was used to “integrate” bulk RNA-seq data from HAECs and the scRNAseq data from plaque endothelium?

Authors could consider rescue experiments to verify if PELP1, AEBP1 and/or COL4A1 are sufficient to induce proliferation or migration in ECs with siTWIST1.

The images of the Twist1ECKO plaques chosen for lesion composition studies are underwhelming (Figure 6). Authors should display BCA sections that are less damaged. Also, Miller’s elastic stain is not conventional to assess plaques size – the reported plaque size is % of what?

Integrating the assessment of atherosclerotic plaques after 8 weeks of high fat diet, prior to tamoxifen, would allow the authors to determine whether knocking out Twist1 leads to lesion regression or stabilized existing plaques.

Reviewer #3

(Remarks to the Author)

This is an interesting paper assessing the role of EC TWIST in experimental and human atherosclerosis. The association with EndMT is addressed and the influence of the development via interventional mouse experiments, human association and mechanistic work.

In the abstract, this is very loose terminology and really correct, “Endothelial-to-mesenchymal transition (EndMT) is associated with advanced atherosclerotic plaques, but its role in plaque progression remains unclear.” Please use more accurate wording to set the scene as there is evidence of causality and it is not “unknown”. Evidence suggests.....yet.....

Additionally in the abstract it states “Twist1 regulates endothelial cell heterogeneity by promoting cell states expressing EndMT markers.” – regulated EC heterogeneity is not proven and is also a loose term. And in Fig 1 b how is heterogeneity proven? Please change this wording. In Figure 1B there are very few cells in effect and the tSNE plot does not really show any differences and is likely over clustered in Figure 1a. eg. Cluster 10 has only 3-4 cells by the looks. What does TWIST itself look like in this plot? The bar graphs are also over interpreted which such low numbers of cells, there are no statistics applied and it is unclear how many replicates are used for the input mouse cells. Overall, Fig 1 shows some changes in the control vs EC TWIST KO but the data is unclear from a statistical point of view, low numbers of cells, thus the interpretation is to be taken with high caution. Additionally, are any of the findings validated? To align the clusters with data back from the mice? And is there an effect of the the loss of EC TWIST one (induced at 8 weeks) on plaque phenotypes? This is assessed a bit later in the manuscript but likely the evidence of mouse functional influence is needed earlier.

In Figure 2 there is the assessment of EndMT in mouse plaques and shows what has been shown before, ie. There are cells that have the lineage tracer in the plaque and they co-expressed ACTA2. This does not seem to be quantified in the data, please add. Also more than one mesenchymal marker is needed. The effect of EC TWIST KO suggests that TWIST KO reduces this. This is quantified as EndMT cells, but please check another marker plus the total number of plaque ECs. The evidence to support the effect of EC TWIST on plaques is well powered but not confirmed with additional experiments, so remains a little weak. Strengthening this would be important. Additionally, what is the endogenous expression of TWIST in the developing plaque vs control at the protein level?

In the DF analysis of the aortic EC, can the authors show the induction of EndMT using a range of markers? It is unclear which markers are used. Also, did a second siRNA show similar effects? Do the genes affected in the human setting also correlate back to the EC KO in vivo aside from PELP and AEBP1 functionally. While it is strong to show the key gene in the eyes of the authors, the broader relevance of the gene set is not followed further. In the TWIST-PELP-1 axis, can PELP1/AEBP1 manipulation rescue the loss of TWIST1 on functional parameters? What we are lacking is key evidence of the importance of the precise axis identified. If strengthened this would have a positive effect on the study.

The human studies could be strengthened in terms of TWIST expression and correlation with EndMT markers. This is a weaker aspect of the paper and needs to be extended with additional markers and quantifications.

Version 1:

Reviewer comments:

Reviewer #1

(Remarks to the Author)

In this revision, the authors have addressed the reviewers' concerns and satisfactorily revised the manuscript.

Reviewer #2

(Remarks to the Author)

The authors have responded appropriately to my queries. Excellent work

Reviewer #3

(Remarks to the Author)

The authors have completed a thorough revision

REVIEWER COMMENTS

Reviewer #1 (Remarks to the Author):

The role of EndMT in atherosclerosis is important, but the mechanisms are still unclear. Previously, Dr. Evans and colleagues showed that a constitutional EC-targeted TWIST1 knockout reduces atherosclerotic plaque development. The current study further reports that TWIST1-driven EndMT plays an unexpected, plaque-stabilizing role in advanced atherosclerosis. Mechanistic studies show that TWIST1-dependent EndMT enhances endothelial migration via PELP1 and activates an AEBP1-COL4A1 pathway to drive endothelial proliferation. These results challenge the prevailing view that EndMT destabilizes plaques. Instead, it suggests a novel concept that TWIST1-driven EndMT promotes plaque stability, reducing rupture risk. Overall, the results support the conclusions. However, there are some concerns.

Specific Concerns:

Figure 6: For this study, the mice were treated with Tamoxifen for 5 days to induce TWIST1 ECKO at an 8-week time-point, followed by an additional 6-week Western diet. Were Control groups also treated with Tamoxifen to account for its potential atheroprotective complications?

This is an important point. We confirm that both control and experimental groups of mice were treated with tamoxifen, and this is now stated clearly in the Methods as follows:

“Experimental and control groups were treated by intraperitoneal administration of tamoxifen (Sigma-Aldrich) in corn oil for five consecutive days (2 mg per mouse per day), followed by Western diet for 6 weeks”. Page 11, paragraph 1.

Did the authors also examine other atheroprone regions, including the aortic sinus and aortic arch? If so, were the results consistent in those regions as well?

We appreciate the reviewer’s insightful comment. To address this point, we carried out new analyses of aortic root plaques, in addition to the brachiocephalic artery, to directly test whether our findings were consistent across different plaque regions. The results were closely consistent between these regions: *Twist1^{ECKO}* mice exhibited smaller plaques, reduced collagen content, fewer ACTA2-positive cells, increased Mac3-positive cells, and greater necrotic core areas in aortic root plaques. These findings are presented in Fig. 7 and described on page 7, paragraph 2. They corroborate our observations in the brachiocephalic artery, further supporting that *Twist1* promotes plaque growth while limiting features associated with plaque instability.

Figure 7: TWIST1 protein expression in human carotid artery sections (n=3) is too small to support the authors' conclusion. The n number should be significantly increased to test the hypothesis.

We agree entirely with the reviewer’s concerns. Analysis of plaques from 78 individuals revealed that *TWIST1* mRNA expression is associated with asymptomatic disease and improved outcome. In response to the reviewer’s understandable concerns, we worked hard to improve our analysis of TWIST1 at the protein level by immunostaining. However, this was challenging because endothelium is incomplete in many samples making accurate measurement of the proportion of TWIST1+ cells difficult. We painstakingly identified symptomatic (n=10) and asymptomatic (n=9) cases with the most intact endothelium for analysis of TWIST1 by immunohistochemistry. We concluded that TWIST1 is expressed in human plaque endothelium but the attempt to

correlate with patient symptoms is likely underpowered because of low numbers. Overall, we can conclude that TWIST1 is expressed in human plaque endothelium but our results remain inconclusive regarding its association with disease severity. These data now appear in Fig S1 and we have described them and alluded to their exploratory nature as follows:

“scRNAseq analysis of murine plaques revealed Twist1 expression in multiple cell types with high levels in EC (Fig. S1A). Analysis of human carotid artery plaques revealed that TWIST1 mRNA levels were enhanced in plaques from asymptomatic compared to symptomatic patients (Fig. S1B), and that higher TWIST1 expression was associated with a reduced risk of future cardiovascular events during post-operative follow-up compared to low levels (Fig. S1C). Immunohistochemistry confirmed TWIST1 protein expression in ECs of human carotid plaques (Fig. S1D) but this exploratory analysis did not reveal differences between symptomatic and asymptomatic groups. It was concluded that TWIST1 is expressed in plaque endothelium and that its potential role in plaque progression should be investigated”.
Page 4, paragraph 1.

Minor: The sources of mice used to generate Twist1 ECKO ApoE^{-/-} and Twist1 ECKO-ApoE on an EC-fate-mapped background should be clarified.
These details have been incorporated into the Methods section of the revised manuscript (page 10, paragraph 1).

Reviewer #2 (Remarks to the Author):

During endothelial-to-mesenchymal transition, ECs lose markers like VE-cadherin and acquire mesenchymal markers such as ACTA2. TWIST1 drives EndMT and promotes lipid accumulation in early atherosclerosis. Unexpectedly, the authors find that TWIST1 has a stabilizing effect on atherosclerotic plaques. An interesting relationship is demonstrated between clusters regarding level of Twist1 promotion and distribution of DF and UF markers, although the relationship with EndMT enrichment clusters is not as striking. The authors explore RNA expression in ECs and identify a set of differently expressed genes from which 10 candidates are prioritized for further assessment based on their TWIST1-dependent regulation. Among those, PELP1, AEBP1 and COL4A1 are singled out as the genes that regulated both proliferation and migration of ECs. Finally, the impact of EC Twist1 KO on atherogenesis is tested in male and female mice, revealing important sex differences. Interestingly, Twist1 ECKO leads to smaller plaques that feature a less stable phenotype, in accord with human data showing that reduced TWIST1 mRNA expression is associated with a greater risk of future cardiovascular events in patients. Overall, the study design is sound and the results convincing.

Comments:

Figure 2B show extensive Cdh5⁺ cells throughout the atherosclerotic lesions, presumably because of Cre recombinase reporter gene expression in haematopoietic cells. Is this representative? If so, how then are these cells accounted for in the quantification of EndMT in Figure 2C? Also, how Twist1 in these cells may contribute to lesion formations should be considered.

We thank the reviewer for raising this important point. In our model, *Cdh5-Cre* is activated in adult mice in response to tamoxifen, thereby avoiding recombination in

fetal hemogenic endothelium. It is therefore highly unlikely that any TdTomato⁺ cells represent a hematopoietic lineage. To track endothelial-derived cells, we induced TdTomato expression in mice with pre-existing plaques and analyzed cell distribution following six weeks of plaque progression. We observed that TdTomato⁺ cells accumulated within plaques at levels ranging from 1–14% in control mice, and that *Twist1* endothelial deletion significantly reduced this accumulation (Fig. 2b). In the revised manuscript, we have substantially improved these data by co-staining TdTomato⁺ cells with markers of EndMT; the majority of TdTomato⁺ cells expressed NCAD and SM22 α (Fig. 2c, Fig. 2d), supporting the conclusion that *Twist1* positively regulates EndMT during murine atherosclerosis. This is described on page 5, paragraph 2 of the revised manuscript.

The authors stipulate that effects of *Twist1* on migration and proliferation primarily occur in DF cells but effects of *Twist1*KO on the function of ECs under UF should also be tested to confirm.

We have revised the manuscript to include an analysis of the potential effects of *TWIST1* silencing in cells exposed to UF. Our results indicate that silencing *TWIST1* shows a trend toward reduced migration and proliferation; however, these effects were not statistically significant. These findings have been incorporated into the revised manuscript (Fig. S12).

What approach was used to “integrate” bulk RNA-seq data from HAECs and the scRNAseq data from plaque endothelium?

The term ‘integrate’ may be misleading, as it could imply a bioinformatics-based dataset integration. Instead, our approach involved identifying genes regulated by *TWIST1* in both datasets and focusing on those that overlapped, i.e. genes regulated by *TWIST1* *in vivo* (plaque endothelium, scRNA-seq, *Twist1*^{ECKO} vs. controls) and *in vitro* (HAECs, bulk RNA-seq, *TWIST1* siRNA vs. controls). We have clarified this wording in the revised manuscript as follows:

“We took a structured approach to define the mechanisms underlying TWIST1-driven migration and proliferation. Firstly, we compared bulk RNA-seq data from HAECs and scRNAseq data from plaque endothelium and identified 45 genes positively regulated by TWIST1 across both systems (Fig. 3h)”. (page 5, paragraph 4).

Authors could consider rescue experiments to verify if *PELP1*, *AEBP1* and/or *COL4A1* are sufficient to induce proliferation or migration in ECs with si*TWIST1*.

We thank the reviewer for this excellent suggestion. To address this, we generated lentiviral constructs to rescue the expression of *PELP1*, *AEBP1*, and *COL4A1* in *TWIST1*-silenced ECs. We found that *PELP1* expression restored migration in *TWIST1*-silenced cells (Fig. S14), while *PELP1*, *AEBP1*, and *COL4A1* each rescued proliferation in these cells (Fig. S15). These findings support the concept that *PELP1*, *AEBP1*, and *COL4A1* function downstream of *TWIST1* to regulate EC migration and proliferation. These data are described in the revised manuscript (page 6, paragraphs 2 and 3).

The images of the *Twist1*ECKO plaques chosen for lesion composition studies are underwhelming (Figure 6). Authors should display BCA sections that are less damaged. We agree and have presented better quality plaque images in the revised manuscript.

Also, Miller's elastic stain is not conventional to assess plaque size – the reported plaque size is % of what?

We apologize that our original description of plaque quantification was not sufficiently clear. Miller's elastin stain provides excellent definition of the internal elastic lamina, enabling accurate delineation of the intimal boundary and thus precise measurement of plaque and intimal areas in murine arterial sections. Elastic fibers stain dark blue to black, which allows clear discrimination of the internal elastic lamina from surrounding structures – see image below. This approach has been used by several laboratories, including recent studies from highly experienced laboratories (PMID: 38881440; PMID: 39276419).

Plaque size was expressed as the percentage of the intimal area occupied by plaque. To confirm the robustness of this method, we also independently quantified plaque area using hematoxylin and eosin (H&E) staining, and these data are now included in the revised manuscript. We have revised the Methods section accordingly to clarify these details as follows:

“Mice were euthanized and perfusion-fixed via the left ventricle with phosphate-buffered saline (PBS), followed by 4% paraformaldehyde (PFA). The brachiocephalic artery and aortic sinus were dissected, carefully cleared of adventitial tissue, and embedded in paraffin. Paraffin blocks were sectioned at 5 μ m thickness. Sections were dewaxed in xylene and rehydrated through graded alcohols using standard protocols prior to histological analysis. Elastic lamina morphology and plaque quantification were assessed using Miller's elastin stain. The internal elastic lamina was delineated, and the areas of the intima and plaque were quantified using ImageJ. Plaque burden was expressed as the percentage of the intimal area occupied by plaque. Hematoxylin and eosin (H&E) staining was used to measure the area of necrotic, acellular material within the plaque, expressed as a percentage of total plaque area. The proportion of plaque area within the intima was also independently verified using H&E staining. Collagen content was assessed using Picrosirius Red staining and expressed as the percentage of plaque area occupied by collagen. The proportions of vascular smooth muscle cells (VSMCs; anti-ACTA2) and macrophages (anti-MAC3; antibodies listed in Table S2) were determined by immunohistochemistry and expressed as percentages of plaque area. For antigen retrieval, sections were boiled in citrate buffer for 10 min. Breaks in elastin fibers within the intima adjacent to plaques were manually counted. All sections were imaged using a Leica Thunder microscope, and quantitative analyses were performed using ImageJ”. Page 10, paragraph 2.

Integrating the assessment of atherosclerotic plaques after 8 weeks of high fat diet, prior to tamoxifen, would allow the authors to determine whether knocking out Twist1 leads to lesion regression or stabilized existing plaques.

We thank the reviewer for this excellent suggestion. As recommended, we exposed *Twist1*^{+/+}*ApoE*^{-/-} mice to a Western diet for 8 weeks and analyzed brachiocephalic (Fig. S20) and aortic root (Fig. S21) atherosclerotic plaques. These measurements served as a baseline for subsequent analysis of plaque progression from 8 to 14 weeks in control versus *Twist1*^{ECKO} mice. This comparison has enhanced our interpretation of *Twist1* biology; it demonstrated that *Twist1* endothelial deletion reduced plaque progression between 8 and 14 weeks but did not induce regression of existing lesions. Moreover, *Twist1* deletion at the 8-week time point enhanced the progression of features associated with plaque instability including increased macrophage content and necrotic core area, while reducing features of stability, such as collagen content and ACTA2-positive cells. These new data are described in the revised manuscript (page 7, paragraph 2).

Reviewer #3 (Remarks to the Author):

This is an interesting paper assessing the role of EC TWIST in experimental and human atherosclerosis. The association with EndMT is addressed and the influence of the development via interventional mouse experiments, human association and mechanistic work.

In the abstract, this is very loose terminology and really correct, “Endothelial-to-mesenchymal transition (EndMT) is associated with advanced atherosclerotic plaques, but its role in plaque progression remains unclear.” Please use more accurate wording to set the scene as there is evidence of causality and it is not “unknown”. Evidence suggests.....yet.....

We agree with the reviewer and have revised the abstract to more accurately reflect the evidence supporting EndMT in plaque progression, as follows: “*Endothelial-to-mesenchymal transition (EndMT) is associated with advanced atherosclerotic plaques, and accumulating evidence suggests that it contributes to plaque progression. To investigate this relationship, we examined the role of TWIST1, a key EndMT-driving transcription factor, in plaque development*”.

Additionally in the abstract it states “Twist1 regulates endothelial cell heterogeneity by promoting cell states expressing EndMT markers” – regulated EC heterogeneity is not proven and is also a loose term. And in Fig 1 b how is heterogeneity proven? Please change this wording.

We agree and have revised the abstract as follows: “*Single-cell RNA sequencing of atherosclerotic plaques from hypercholesterolemic mice with inducible deletion of Twist1 from endothelial cells (Twist1^{ECKO} ApoE^{-/-}) revealed that Twist1 drives the emergence of endothelial cell subsets characterized by EndMT-associated gene expression*”.

In Figure 1B there are very few cells in effect and the tSNE plot does not really show any differences and is likely over clustered in Figure 1a. eg. Cluster 10 has only 3-4 cells by the looks.

We appreciate the reviewer’s comment regarding the clustering and t-SNE visualization. Clustering resolution in scRNA-seq analyses is typically set between 0.4

and 1.4, with 0.8 being the commonly used default. To ensure that our clustering was biologically meaningful and not over-segmented, we performed iterative clustering across multiple resolutions (0.4–1.0) with expert guidance from Joanna Kalucka and Helle Jorgensen – please see figure below. At a resolution of 0.5, cluster 3 contained groups of cells that exhibited distinct gene signatures that were appropriately separated at resolution 0.8 (becoming clusters 4 and 7). Similarly, cluster 1 at resolution 0.5 included cells with disparate transcriptional identities, which were resolved at resolution 0.8 (becoming clusters 2 and 3). On this basis, we selected 0.8 as the optimal resolution for this dataset.

For a single-cell dataset focused specifically on endothelial populations, our sort-seq analysis of ~2000 cells is consistent with published studies of comparable scope. While cluster 10 contains relatively few cells, it remained a stable and distinct cluster across all resolutions examined due to its unique gene expression profile. Additional analyses confirmed that this cluster represented contaminating smooth muscle cells, and we therefore excluded it from downstream endothelial-focused analyses (Fig. S8; page 4, paragraph 4).

What does TWIST itself look like in this plot?

We thank the reviewer for this helpful question. We have revised the manuscript to include the expression pattern of *Twist1* in the updated figure (Fig. S5). *Twist1* is expressed broadly across endothelial cell subsets, with a clear enrichment in subset 7, which is consistent with this subset being prominently regulated by *Twist1* (described on page 4, paragraph 3).

The bar graphs are also over interpreted which such low numbers of cells, there are no statistics applied and it is unclear how many replicates are used for the input mouse cells. Overall, Fig 1 shows some changes in the control vs EC TWIST KO but the data is unclear from a statistical point of view, low numbers of cells, thus the interpretation is to be taken with high caution. Additionally, are any of the findings validated? To align the clusters with data back from the mice?

We thank the reviewer for these valuable comments. In our study, endothelial cells were pooled from multiple mice (*Twist1*^{ECKO}, N=4; controls, N=5), yielding approximately 2000 endothelial cells for scRNA-seq analysis. This number provides adequate resolution to detect transcriptional differences across endothelial subpopulations. In Fig. 1, we present these data as violin plots enabling visualization

of the distribution within each cluster. The pooled data suggest that *Twist1* positively regulates clusters 2, 7, and 8.

We fully acknowledge the reviewer’s concern regarding statistical interpretation. Statistical testing cannot be applied directly to pooled single-cell data because individual cells are not independent biological samples. To address this, we performed a complementary analysis using mouse-level aggregated values - data presented below for the reviewer’s reference. This analysis reveals trends consistent with the pooled dataset and makes the inter-mouse variability explicit. Two-way ANOVA demonstrated borderline significance for the effects of *Twist1*^{ECKO} on clusters 2 and 8, reflecting the biological variation between mice.

Given these considerations, we believe the clearest and most appropriate way to present the scRNA-seq data is through violin plots of data pooled from multiple mice, while explicitly stating that the analysis is exploratory. We have revised the manuscript accordingly to read:

*“An exploratory analysis of pooled data from *Twist1*^{ECKO} *ApoE*^{-/-} mice and control mice suggested that clusters 2, 7, and 8 were enriched in cells from control mice, whereas clusters 4 and 5 contained a higher proportion of cells from *Twist1*^{ECKO} *ApoE*^{-/-} mice (Fig. 1b; Fig. S4a; Fig. 1c).”*

Importantly, our scRNAseq analysis was informative since its major findings were independently validated *in vivo*. scRNAseq analysis suggested that *Twist1* regulates EndMT and this was subsequently confirmed by lineage tracing coupled to immunostaining for the EndMT markers NCAD and SM22A (Fig. 2). In addition, we validated predictions related to cluster 8 by showing that *Twist1* regulates the expression of PELP1 (Fig. 4), AEBP1, and COL4A1 (Fig. 5) which are enriched in cluster 8 (Fig. S5). These data provide independent support for the relevance of cluster 8 and the biological pathways inferred from the scRNA-seq analysis. We describe this in the manuscript (page 7, paragraph 1) as follows:

*“Our data support the suggestion that *Twist1* regulates cluster 8, which is enriched for *Pelp1*, *Aebp1*, and *Col4a1*, and they are consistent with a TWIST1–AEBP1–COL4A1 axis that promotes endothelial cell proliferation.”*

Together, these validation experiments show that the exploratory scRNA-seq analysis generated biologically meaningful hypotheses that were substantiated through independent *in vivo* studies.

And is there an effect of the loss of EC TWIST one (induced at 8 weeks) on plaque phenotypes? This is assessed a bit later in the manuscript but likely the evidence of mouse functional influence is needed earlier.

We thank the reviewer for this thoughtful comment. We carefully considered the positioning of the plaque phenotype data within the manuscript. Our intent was to present the findings in the sequence in which the biological insights emerged—first identifying *Twist1* as a regulator of EndMT in atherosclerosis, and then examining the functional consequences on plaque development. This progression reflects the logic of our discovery process and, in our view, provides a clear narrative for how the mechanistic observations informed the subsequent *in vivo* phenotypic analyses.

In Figure 2 there is the assessment of EndMT in mouse plaques and shows what has been shown before, ie. There are cells that have the lineage tracer in the plaque and they co-express ACTA2. This does not seem to be quantified in the data, please add. Also, more than one mesenchymal marker is needed. The effect of EC TWIST KO suggests that TWIST KO reduces this. This is quantified as EndMT cells, but please check another marker plus the total number of plaque ECs.

We appreciate the reviewer's thoughtful comment and agree that additional validation using multiple mesenchymal markers was necessary. To address this, we performed further immunofluorescence analyses in EC-tracked mice to assess whether EC-derived cells within the plaque co-express established EndMT markers. Specifically, we examined NCAD and SM22 and confirmed that tdTomato+ EC-derived plaque cells express both markers (Fig. 2c, Fig. 2d). We quantified these populations and found that the accumulation of tdTomato+/NCAD+ and tdTomato+/SM22+ in plaques is reduced in *Twist1^{ECKO}* mice compared with controls. These additional data strengthen our conclusion that endothelial *Twist1* promotes EndMT within atherosclerotic plaques.

The evidence to support the effect of EC TWIST on plaques is well powered but not confirmed with additional experiments, so remains a little weak. Strengthening this would be important.

We appreciate the reviewer's comment that our original analysis of plaques was well powered but would benefit from additional confirmation. In the original manuscript, we reported the effects of endothelial *Twist1* on brachiocephalic artery plaques. To strengthen these findings, we now include new data assessing the effects of endothelial *Twist1* deletion on aortic root plaques. These new analyses show consistent effects across vascular regions: *Twist1^{ECKO}* mice exhibit smaller aortic root plaques with reduced collagen content, fewer ACTA2-positive cells, increased Mac3-positive cells, and larger necrotic core areas in the aortic sinus (Fig. 7), which mirror the effects observed in brachiocephalic artery plaques (Fig. 6). Collectively, these data from two independent vascular sites provide strong support for the conclusion that endothelial *Twist1* promotes plaque growth while limiting features associated with plaque instability.

Additionally, what is the endogenous expression of TWIST in the developing plaque vs control at the protein level?

To address this, we attempted to assess TWIST1 protein expression in murine plaque sections by immunostaining. We tested five commercially available antibodies but these experiments were technically challenging due to non-specific binding. Therefore, as an alternative, we analyzed published scRNA-seq datasets to assess the cellular distribution of *Twist1* within plaques. These data, which show that *Twist1* is expressed in multiple cell types including endothelial cells, are presented in Fig. S1a.

In the DF analysis of the aortic EC, can the authors show the induction of EndMT using a range of markers? It is unclear which markers are used.

We now present the effects of *TWIST1* siRNA on individual EndMT markers as a heatmap in Fig. S11c.

Also, did a second siRNA show similar effects?

We agree that using a second siRNA is an important validation. To address this, we repeated our experiments with a second siRNA, *siTWIST1(v2)*, and confirmed that silencing *TWIST1* reduced markers of EndMT at both the mRNA (*SNAI1*, *ACTA2*, *NCAD*; Fig. 3c) and protein (*SNAI1*, *ACTA2*; Fig. 3d) levels. This is described in the manuscript (page 5, paragraph 3) as follows:

“To validate, silencing was carried out using siTWIST1(v2), which reduced EndMT markers by qRT-PCR (SNAI1, ACTA2, NCAD; Fig. 3c) and immunofluorescence (SNAI1, ACTA2; Fig. 3d).”

We further demonstrated that *siTWIST1(v2)* reduces endothelial cell migration (Fig. S14) and proliferation (Fig. S15), reinforcing the conclusion that *TWIST1* is a positive regulator of these processes. This is described on page 6, paragraph 2:

“Silencing using siTWIST1(v2) reduced migration (Fig. S14) and proliferation (Fig. S15), thereby validating TWIST1 as a positive regulator.”

Do the genes affected in the human setting also correlate back to the EC KO in vivo aside from PELP and AEBP1 functionally. While it is strong to show the key gene in the eyes of the authors, the broader relevance of the gene set is not followed further. To address the broader relevance of *TWIST1*-regulated genes, we took a structured, stepwise approach to identify genes consistently regulated by *TWIST1* both *in vivo* and *in vitro*. Comparison of our datasets identified 45 genes regulated by *TWIST1* in both settings. We then performed literature searches and prioritized 24 genes with putative roles in EMT or EndMT for potential functional studies. We validated *TWIST1*-dependent regulation by qRT-PCR in cultured arterial ECs and identified 10 genes that were consistently regulated. Each was individually silenced, and effects on EC migration and proliferation were assessed (Fig. S13c). Seven genes affected proliferation, and one gene (*PELP1*) regulated both proliferation and migration. Because detailed biochemical analysis of all 10 genes was not feasible, we prioritized *PELP1* for its dual functional role and *AEBP1* and *COL4A1* because they form a known signalling axis in other contexts. We have revised the manuscript to clarify this stepwise process (Fig. S13; page 5, paragraph 4 to page 6, paragraph 1). This approach demonstrates the broader relevance of the *TWIST1*-regulated gene set while focussing mechanistic analyses on *PELP1* and the *AEBP1*–*COL4A1* axis.

In the *TWIST1*-*PELP1* axis, can *PELP1*/*AEBP1* manipulation rescue the loss of *TWIST1* on functional parameters? What we are lacking is key evidence of the

importance of the precise axis identified. If strengthened this would have a positive effect on the study.

We thank the reviewer for this comment. To address it, we generated lentiviral constructs to rescue the expression of PELP1, AEBP1, and COL4A1 in TWIST1-silenced ECs. We found that PELP1 expression restored migration in TWIST1-silenced cells (Fig. S14), while PELP1, AEBP1, and COL4A1 each rescued proliferation in these cells (Fig. S15). These findings support the concept that PELP1, AEBP1, and COL4A1 function downstream of TWIST1 to regulate EC migration and proliferation. These data are described in the revised manuscript (page 6, paragraphs 2 and 3).

The human studies could be strengthened in terms of TWIST expression and correlation with EndMT markers. This is a weaker aspect of the paper and needs to be extended with additional markers and quantifications.

We agree with the reviewer. We analysed TWIST1 in human plaques to assess relevance to human disease. Analysis of TWIST1 mRNA in plaques from 78 individuals revealed that TWIST1 expression is associated with asymptomatic disease and improved outcome. In response to the reviewer's understandable concerns, we worked hard to improve our analysis of TWIST1 at the protein level by immunostaining. However, this was challenging because endothelium is incomplete in many samples making accurate measurement of the proportion of TWIST1+ cells difficult. We painstakingly identified symptomatic (n=10) and asymptomatic (n=9) cases with the most intact endothelium for analysis of TWIST1 by immunohistochemistry. We concluded that TWIST1 is expressed in human plaque endothelium but the attempt to correlate with patient symptoms is likely underpowered because of low numbers. Overall, we can conclude that TWIST1 is expressed in human plaque endothelium but our results remain inconclusive regarding its association with disease severity. These data now appear in Fig S1 and we have described them and alluded to their exploratory nature as follows:

“scRNAseq analysis of murine plaques revealed Twist1 expression in multiple cell types with high levels in EC (Fig. S1A). Analysis of human carotid artery plaques revealed that TWIST1 mRNA levels were enhanced in plaques from asymptomatic compared to symptomatic patients (Fig. S1B), and that higher TWIST1 expression was associated with a reduced risk of future cardiovascular events during post-operative follow-up compared to low levels (Fig. S1C). Immunohistochemistry confirmed TWIST1 protein expression in ECs of human carotid plaques (Fig. S1D) but this exploratory analysis did not reveal differences between symptomatic and asymptomatic groups. It was concluded that TWIST1 is expressed in plaque endothelium and that its potential role in plaque progression should be investigated” (Page 4, paragraph 1).

NCOMMS-25-29683A

We were delighted that the referees appreciated our revisions and recommend acceptance of this paper. We are very grateful for their excellent feedback which has substantially improved our paper.